



# Reconstructing post-Jurassic overburden in Central Europe: New insights from mudstone compaction and thermal history analyses of the Franconian Alb, SE Germany

Simon Freitag[1], Michael Drews[2], Wolfgang Bauer[1], Florian Duschl[2], David Misch[3], Harald Stollhofen[1]

[1]GeoZentrum Nordbayern, Friedrich-Alexander University (FAU) Erlangen-Nürnberg, Schlossgarten 5, 91054 Erlangen, Germany
[2]Geothermal Technologies, Technical University of Munich (TUM), Arcisstraße 21, 80333 Munich, Germany
[3]Department für Angewandte Geowissenschaften und Geophysik, Montanuniversität Leoben, Peter-Tunner-Straße 5, 8700 Leoben, Austria

*Correspondence to:* Simon Freitag (simon.s.freitag@fau.de)

**Abstract**

The Franconian Alb of SE Germany is characterized by large-scale exposures of Jurassic shallow marine limestones and dolostones which are frequently considered as outcrop analogues for deep geothermal reservoir rocks in the North Alpine Foreland Basin farther south. However, the burial history of the Franconian Alb Jurassic strata is not well known as they were affected by emersion, leading to extensive erosion and karstification with only remnants of the original Cretaceous and Cenozoic cover rocks preserved. To estimate the original thicknesses of the post-Jurassic overburden we investigated the petrophysical properties and the thermal history of Lower and Middle Jurassic mudstones to constrain their burial history in the Franconian Alb area. We measured mudstone porosities, densities, and maturities of organic material and collected interval velocities from seismic refraction and logging data in shallow mudstone-rich strata. Mudstone porosities and P-wave velocities vertical to bedding were then related to a normal compaction trend that was calibrated on stratigraphic equivalent units in the North Alpine Foreland Basin. Our results suggest maximum burial depths of 900 - 1700 m of which 300 - 1100 m are attributed to Cretaceous and younger sedimentary rocks overlying the Franconian Alb Jurassic units. Compared to previous considerations this implies a more widespread distribution and increased thicknesses of up to ~900 m for Cretaceous and up to ~200 m for Cenozoic units in SE Germany. Maximum overburden is critical to understand mechanical and diagentical compaction of the dolostones and limestones of the Upper Jurassic of the Franconian Alb. The results of this study therefore help to better correlate the deep geothermal reservoir properties of the Upper Jurassic from outcrop to reservoir conditions below the North Alpine Foreland Basin. Here, the Upper Jurassic geothermal reservoir can be found at depths of up to 5000 m.


# 1 Introduction

## 1.1 Palaeogeographic framework

The Franconian Alb east, south and north of the city of Nuremberg (Figure 1) is well known for its impressive
exposures of Jurassic carbonates and reef structures in an area extending for ~120 km east-west and ~160 km
north-south. The area is partly underlain by older basin structures such as the SW-NE trending Carboniferous-
Permian Kraichgau Basin (Lützner and Kowalczyk, 2012) and the Upper Permian-Triassic Franconian Basin
(Freudenberger et al., 2013). Following dominantly terrestrial deposition during the Upper Triassic Keuper, marine
environments returned during the Early Jurassic (Liassic), when the South German Basin was flooded by the
Tethys Ocean, depositing mostly clays and clayey marls (Figure 2) (Piénkowski et al., 2008). Alternating dark
clays and oolitic ironstones then record the Middle Jurassic (Dogger) (Piénkowski et al., 2008). With progressive
shallowing of the epicontinental sea during the Late Jurassic (Malm), massive lime- and marlstone units, including
siliceous sponge-microbial reefs and oolite platforms formed (Koch and Munnecke, 2016; Meyer and Schmidt-
Kaler, 1990; Piénkowski et al., 2008).

The early Cretaceous was characterized by uplift contemporaneous to an overall marine regression leading to
pronounced erosion and karstification of the Franconian Alb Jurassic under tropical to subtropical climates
(Schröder, 1968; Voigt et al., 2007). Uplift of the Bohemian Massif likely amounted up to ~1-1.5 km (Peterek et
al., 1996; Peterek and Schröder, 2010; Reicherter et al., 2008; Schröder, 1987; Wagner et al., 1997), probably
related to far-field compression (Scheck-Wenderoth et al., 2008) and a wrench-dominated tectonic regime at the
southern end of the North Sea rift system (Pharaoh et al., 2010). The uplifted basement areas of the Bohemian
Massif and their eroded sedimentary cover sourced the coarse clastic-terrestrial Schutzfeldschichten (Lower
Cretaceous) which likely covered the entire Franconian Alb (Freudenberger and Schwerd, 1996). Only in the
course of several major northward marine transgressions during the Upper Cretaceous the Franconian Alb area
became flooded and successively buried by a thick pile of mixed siliciclastic and calcareous sediments. The initial
collision between the African and the European plate during the Late Cretaceous then led to widespread inversion
tectonics (Scheck-Wenderoth et al., 2008; Voigt et al., 2008, 2021; Walter, 2007), resulting in the removal of the
majority of Cretaceous sediments. A likely second major uplift phase was induced by the Alpine continental
collision between the latest Late Cretaceous and Palaeocene (Peterek et al., 1997; Reicherter et al., 2008; Schröder,
1987; Wagner et al., 1997; Ziegler, 1987). This, together with mantle-induced domal uplift below the Upper Rhine
Graben Rift to the west of the Franconian Alb area (Figure 1) caused southward tilting of the Mesozoic strata.
Subsequent and tilting-related differential erosion in turn resulted in the characteristic scarpland morphology
(Meschede, 2018; Schröder, 1968; Walter, 2007), leaving only local erosional remnants and residual weathering
products (e.g. Kallmünz boulders, Alblehm) witnessing former Cretaceous overburden (Glaser et al., 2001;
Schirmer, 2015).
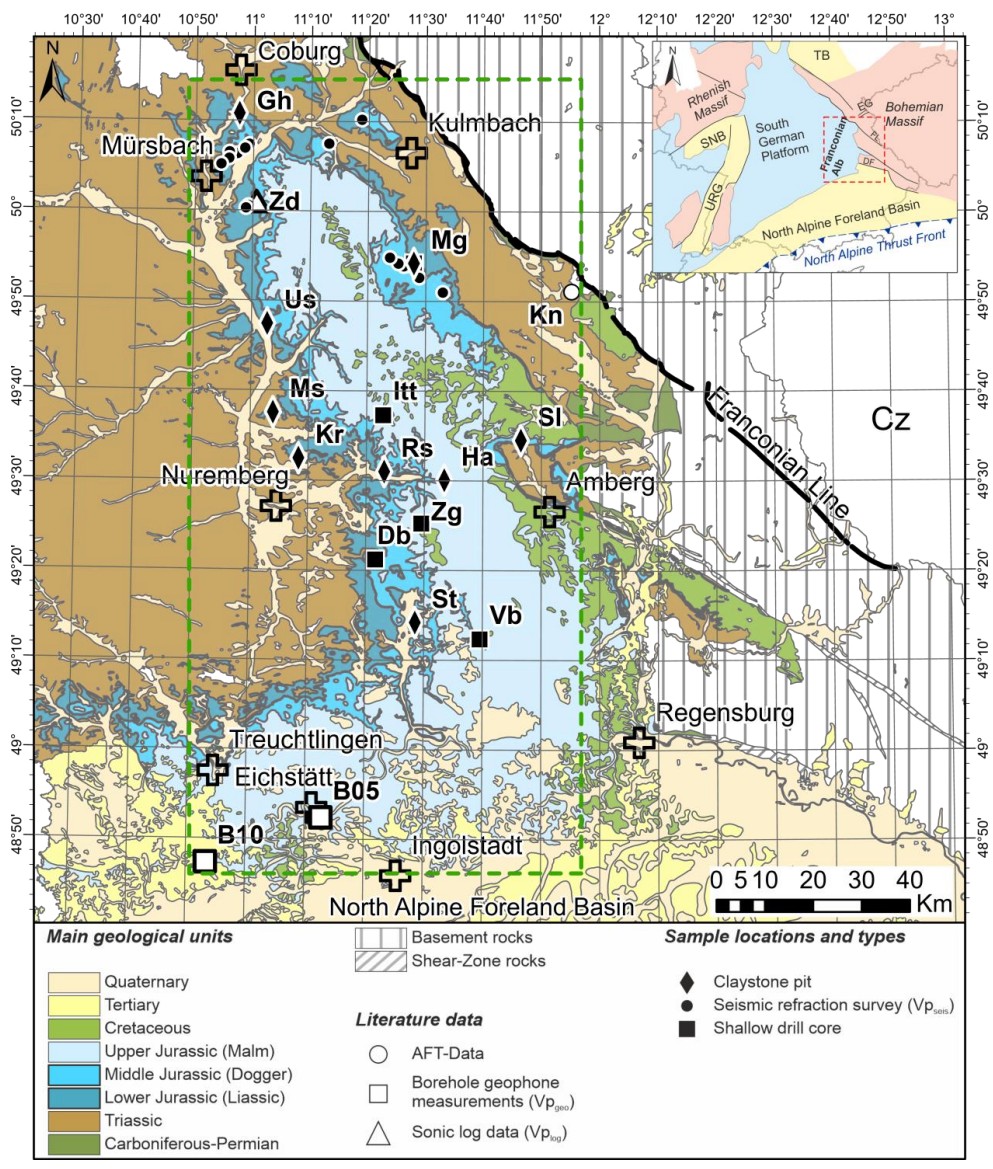

**Figure 1: Geological map, including sample locations and sample types in the Franconian Alb area (green dashed box) with sampling focused on the Lower and Middle Jurassic units (dark blue colour fill). Abbreviations for claystone sample locations: Großheirath (Gh), Hartmannshof (Ha), Kalchreuth (Kr), Mistelgau (Mg), Marloffstein (Ms), Reichenschwand (Rs), Schönlind (Sl), Sengenthal (St), Unterstürmig (Us); abbreviations for seismic refraction data and**

**positions of shallow drill cores: Dörlbach (Db), Ittling (Itt), Mistelgau (Mg), Velburg (Vb), Zankschlag (Zg). Locations of samples used in AFT-studies (white circle, Hejl et al., 1997) are Kemnath (Kn), seismic borehole data (white squares) are from Eichstätt (B05) and Daiting (B10) (Buness and Bram, 2001; Welz, 1994), and sonig log data are from Zapfendorf (Zd) (white triangle, Welz, 1994). Cz = Czech Republic. Inset at upper right shows the location of the study area (red dashed box) in SE Germany and of relevant geological units in neighbouring areas (EG = Eger Graben, SNB**

**= Saar-Nahe-Basin, TB = Thuringian Basin, URG = Upper Rhine Graben). Background data source: Bayerisches Landesamt für Umwelt, www.lfu.bayern.de.**

Following long-lasting denudation, Cenozoic subsidence of the North Alpine Foreland Basin towards the south, contemporaneous to ongoing uplift of basement areas towards the east, led to erosional retreat of incised valleys




that accomodated fluvial clastics during periods of base level rise in the southern Franconian Alb area (Jin et al.,

1995; Meyer, 1996; Zweigel et al., 1998). However, since in the Franconian Alb area only locally a few remnants

of Cretaceous and Cenozoic sediments are preserved (Dill, 1995; Peterek et al., 1997; Peterek and Schröder, 2010),

its post-Jurassic burial history is rather uncertain.

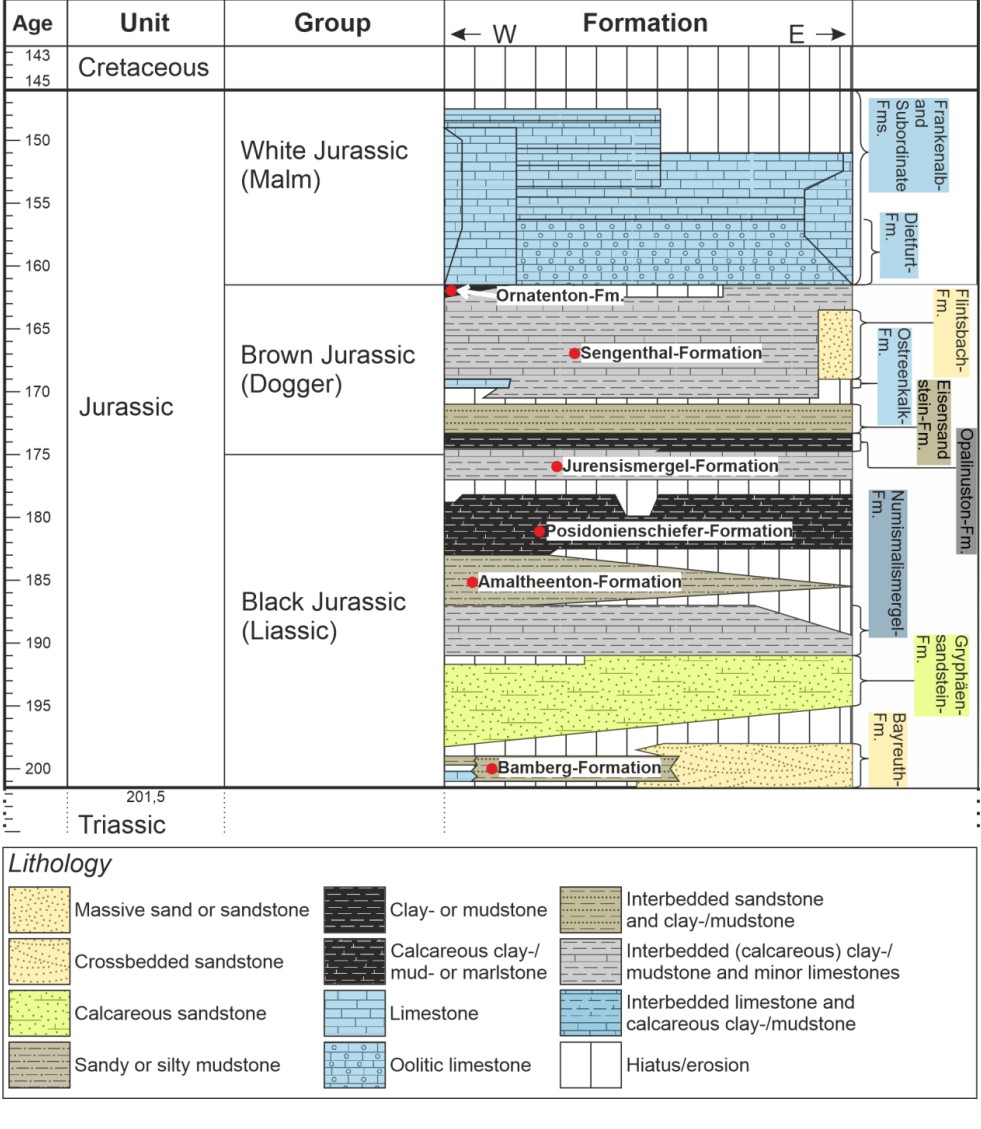

**Figure 2: Jurassic stratigraphy of the Franconian Alb area with the stratigraphic positions of the samples marked by red dots (Modified after German Stratigraphic Commission, 2016).**



### 1.2 Regional post-Jurassic thicknesses

Rather complete records of Cretaceous and Cenozoic sediments are only available within and below the central
and eastern parts of the Cenozoic North Alpine Foreland Basin (Figure 1; NAFB) in SE Germany and Upper
Austria. There, seismic and borehole data based thicknesses of up to 900 m (Przybycin et al., 2015) or even up to
1000 m of Cretaceous (Meyer, 1996; Walter, 2007) and up to 5000 m of Cenozoic sediments are reported
(Bachmann and Müller, 1992). The Franconian Alb area directly north of the North Alpine Foreland Basin
however had a different post-Jurassic, in particular post-Cretaceous burial history with the line Ingolstadt-
Regensburg (Figure 1) roughly dividing areas of Cenozoic subsidence versus non-subsidence and/or uplift.
Towards the north, remnants of Cretaceous strata are only present on the eastern flank of the Franconian Alb, close
to the Franconian Line (Figure 1) (Meyer, 1996), a prominent NW-SE-striking, steeply NE-dipping upthrust fault
that was repeatedly reactivated since the Permo-Triassic and superimposes basement rocks onto the Permo-
Mesozoic sediment cover (Schröder, 1987; Zulauf, 1993). Nevertheless, the areal extent of sediment overburden
since the Cretaceous still remains unclear (Eberle et al., 2017; Niebuhr et al., 2009) and only a few studies (Peterek
and Schröder, 2010; Schröder, 1970, 1987) considered the burial history of the Franconian Alb and the original
thicknesses of post-Jurassic sediments.

Based on geological field observations, Schröder (1970, 1987) estimated an original thickness of >300 m of
Cretaceous sediments in the Franconian Alb area, a value which has later been confirmed by Meyer (1996) and
Peterek and Schröder (2010), based on palaeogeographic considerations. From other published data a rough picture
emerges of a Cretaceous sediment cover decreasing from ~1-2 km (Hejl et al., 1997; Schröder, 1987) directly in
front of the Franconian Line down to about 200-400 m farther west (Meyer, 1996; Niebuhr et al., 2009; Peterek
and Schröder, 2010; Schröder, 1970; Voigt et al., 2008; Walter, 2007), eventually leading to total pinch out towards
the W to SW (Peterek and Schröder, 2010). Hejl et al. (1997) used apatite fission-track (AFT) analysis to determine
the low-temperature history for ortho- and paragneiss boulders that are situated to the east of the Franconian Alb,
close to the Franconian Line. They infer a burial of up to 2000 m for Upper Cretaceous clastics in the proximal
southwestern vicinity of the Franconian Line. Another more comprehensive AFT and (U-Th)/He analysis-based
thermochronological study by von Eynatten et al. (2021) on the exhumation history of central Germany, including
the Franconian Platform, points to large areas of Late Cretaceous to Paleocene domal uplift that experienced
removal of 3-4 km of Mesozoic strata. In contrast, average vitrinite reflectance data of 0.7-0.8% for Lower Keuper
(Ladinian) sediments just west of the northern Franconian Alb area constrain a much lower burial depth of 1.4 km
(Bachmann et al., 2002). Subtracting reported regional Middle/Upper Keuper and Jurassic sediment thicknesses
of 900 m in the southern and 1400 m in the northern Franconian Alb area (Freudenberger and Schwerd, 1996)
would suggest that no or only a <500 m thick post-Jurassic sediment cover was existing. As all of these studies
did not quantify the maximum post-Jurassic sediment overburden, we aim to tackle this question by combining
several methodological approaches that rely on independent data sets.

### 1.3 The burial memory of mudstones

The degree of compaction has a strong influence on the mudstones' petrophysical properties, such as sonic
velocity, density and porosity (Baig et al., 2019; Bjørlykke, 1999; Bjørlykke and Høeg, 1997; Chilingar and
Knight, 1960; Mondol et al., 2008; Vasseur et al., 1995; Yang and Aplin, 2004). Mudstone compaction has been



intensively studied in the past (Aplin et al., 2006; Baig et al., 2019; Bowers, 1995; Dewhurst et al., 1998; Djéran-Maigre et al., 1998; Thyberg and Jahren, 2011; Luo and Vasseur, 1995; Vasseur et al., 1995; Yang and Aplin, 1997, 2004; Yin, 1992) and is mainly controlled by grain size (Fawad et al., 2010; Mondol et al., 2007; Yang and Aplin, 2004), mineralogical composition (Fawad et al., 2010; Marion et al., 1992; Mondol et al., 2007), and texture (Fawad et al., 2010; Marion et al., 1992; Mondol et al., 2007). As the mudstones' compaction behaviour is thought to be almost irreversible even after unloading they are particularly well suited to record maximum burial, respectively overburden (Baig et al., 2019; Henk, 1992; Hillis, 1995; Issler, 1992; Magara, 1976; Mavromatidis and Hillis, 2005). Another source of information for maximum burial of mudstones is given by vitrinite reflectance, a measure of the increasing thermal maturation of organic matter contained in mudstones (Hertle and Littke, 2000; Liu et al., 2020; Sweeney and Burnham, 1990).

### 1.4 Study aim

In this study we combine mudstone porosity and density data from Helium and Mercury porosimetry with vitrinite reflectance data and mudstone velocity data from downhole sonic velocity, downhole geophone and seismic refraction field surveys to gain independent insights on the maximum burial of the Franconian Alb. The results will be compared with and discussed in the context of previous studies (Bader, 2001; Hejl et al., 1997; Peterek and Schröder, 2010; Schröder, 1987; von Eynatten et al., 2021).

Our results shed new light on the evolution of the Franconian Alb area and the original distribution and thicknesses of Cretaceous and Cenozoic sediments in Central Europe. They are also of great relevance for an improved understanding of diagenetic pathways and hydraulic properties of the Permo-Triassic clastics and Late Jurassic carbonate rocks in the Franconian Alb. The latter serve as important outcrop analogue for the most important deep geothermal (Malm) aquifer in the North Alpine Foreland Basin (Kröner et al., 2017; Mraz et al., 2018), whose petrophysical properties are known to strongly depend on burial depth (Bohnsack et al., 2020, 2021; Homuth et al., 2014; Steiner et al., 2014). Finally, the integration of different parameters and measurement types provides an important reference data set (Table A- 2) for future studies, aiming to use petrophysical properties of exhumed and near-surface located mudstones for burial history studies.

## 2 Data and Methods

### 2.1 Franconian Alb sample locations and data sources

We collected Lower (Liassic) and Middle Jurassic (Dogger) clay-/mudstone samples (Figure 2) across the Franconian Alb area along a N-S transect from Coburg to Eichstätt and from Treuchtlingen to Amberg in east-west direction (Figure 1). Table 1 summarizes all sample locations, sample sources, sample types, sample depth below ground, and stratigraphic positions in addition to applied methods and number of measurements per sample.



**Table 1: List of sample locations, sources, types, mean true vertical depth (TVD), stratigraphic unit, applied methods and number of measurements per sample. See Figure 1 for sample locations and Figure 2 for stratigraphic overview.**

| Location | Source | Type | Mean TVD [m] | Stratigraphic unit | GSC | $\rho_t$ | $\rho_b$ / $\emptyset_{hg}$ | Vp | VR | XRD |
|---|---|---|---|---|---|---|---|---|---|---|
| Daiting (B10) | (Buness and Bram, 2001) | Borehole geophone | 455.0 | Dogger | | | | 1 | | |
| Dorfbach (Db) | This study | Core | 6.7 | Posidonienschiefer-Fm. | | 4 | 7 | 2 | | 20 |
| Eichstätt (B05) | (Buness and Bram, 2001) | Borehole geophone | 327.0 | Dogger | | | | 1 | | |
| Großheirath (Gh) | This study | Claystone pit | 0.5 | Bamberg-Fm. | 2 | 1 | 2 | | 1 | |
| Hartmannshof (Ha) | This study | Claystone pit | 0.5 | Sengenthal-Fm. | | 1 | 1 | | 2 | 1 |
| Itting (Itt) | This study | Core | 20.7 | Ornatenton | | 5 | 5 | | | 5 |
| Kalchreuth (Kr) | This study | Claystone pit | 0.0 | Amaltheenton | | 1 | 1 | | 1 | |
| Marloffstein (Ms) | This study | Claystone pit | 0.5 | Amaltheenton | | 1 | 1 | | | 1 |
| Mistelgau (Mg) | This study | Claystone pit & Core | 0.5 | Jurensismergel | 2 | 4 | 7 | | 3 | 7 |
| Reichenschwand (Rs) | This study | Claystone pit | 0.5 | Amaltheenton | | 1 | 1 | | | 1 |
| Northern study area* | This study | Seismic survey | 15.0 - 45.0 | Liassic to Dogger | | | | 40 | | |
| Schönlind (Sl) | This study | Claystone pit | 0.0 | Amaltheenton | 2 | 1 | 2 | | 1 | |
| Sengenthal (St) | This study | Claystone pit | 0.5 | Sengenthal-Fm. | 1 | 1 | 2 | | | 1 |
| Unterstürmig (Üs) | This study | Claystone pit | 0 | Amaltheenton | 3 | 3 | 2 | | | |
| Velburg (Vb) | ABDNB | Core | 41.1 | Eisensandstein- to Sengenthal-Fm. | 3 | 3 | 14 | | 1 | |
| Zankschlag (Zs) | ABDNB | Core | 57.1 | Sengenthal-Fm. | 27 | 8 | 27 | | | |
| Zapfendorf (Zd) | (Welz, 1994) | Sonic Log | 20.5 | Amaltheenton- to Jurensismergel-Fm. | | | | 22 | | |

* Refraction velocities for low velocity layers from a refraction seismic survey (see Figure 1): Messenfeld - Unternsdorf (N of Bad Staffelstein); Hohengüßbach - Wildenberg ( E of Breunach to S of Kronach); Draisdorf - Gotsfeld (W of Bad Staffelstein to W of Creußen).

Abbreviations: ABDNB = Autobahndirektion Nordbayern; GSC = grain size classification; $\rho_t$ = oure (skeletal) density; $\rho_b$ = Bulk density; Vp = p-wave velocity (in situ); VR = Vitrinite reflectance; XRD = X-ray diffraction.


Measured and calculated values for each sample are shown in Appendix Table A- 1. Macroscopically "pure" Jurassic clay-/mudstones (minimum sample size 10 x 10 x 10 cm) were selectively sampled at 0.5 m minimum depth (to avoid alteration/weathering) from nine active and closed claystone pits and from five newly drilled shallow drill cores (up to 12 m below ground level). Except for core samples from Velburg and Zankschlag all samples were packed and stored in an air-evacuated light-, water- and air-proof aluminium barrier foil directly after extraction to preserve the best possible *in-situ* conditions. Interval velocity data of Lias and Dogger clay-/mudstones from a shallow seismic refraction survey for low velocity layers in the course of this study (see Figure 1 for locations), published borehole geophone data of Buness and Bram (2001) and sonic log velocity data from a shallow wellbore (Zapfendorf) in the NW part of the study area (Welz, 1994) were also integrated.

**2.2    Reference data from the North Alpine Foreland Basin**

Density and sonic log data of 9 deep wells in the North Alpine Foreland Basin (Figure 3) have been filtered for appropriate mudstone intervals using gamma-ray (mudstone cut-off at 60-120 API) and/or resistivity values (mudstone cut-off at 4-8 Ωm) as a mudstone discriminator and log values were

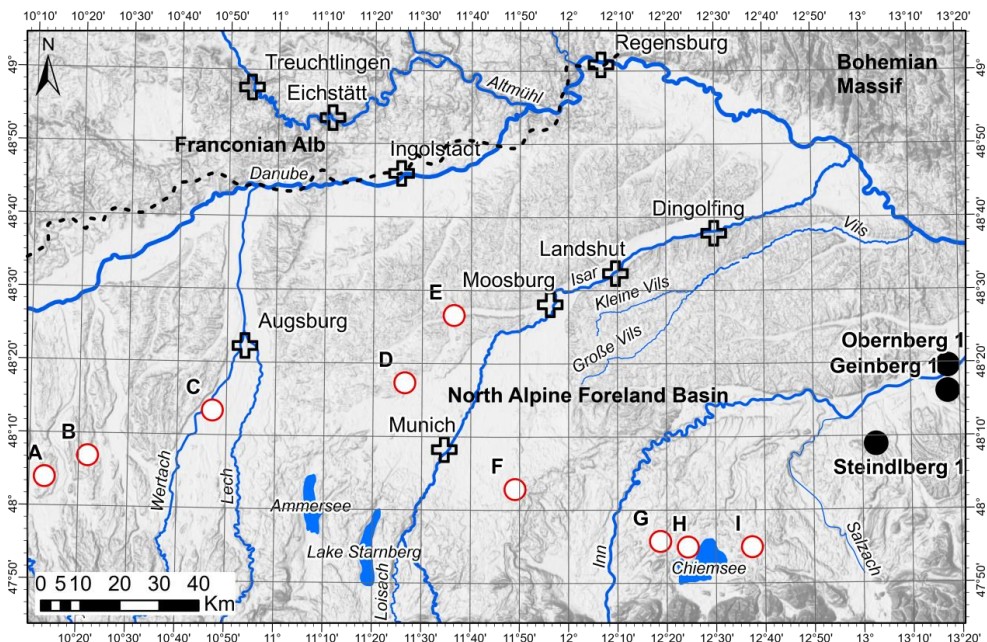

**Figure 3: Map of the North Alpine Foreland Basin just south of our study area (c.f., Figure 1). Bold black dashed line in the Danube River area indicates the present day erosional edge of the North Alpine Foreland Basin fill, based on Bachmann and Müller (1992). White dots with red rims represent (anonymised) well locations of which density and sonic log data are used in this study; black dots indicate well locations which were sampled for Vitrinite reflectance (VR) measurements; white crosses mark larger cities. Modern drainage systems and lake bodies are highlighted in blue. Background lake- and river-data were provided by the European Environment Agency (EEA; status: published 23 Feb 2009, last modified 29 Nov 2012; downloaded 19 July 2021 at 12:36) and the "Bundesanstalt für Gewässerkunde" (WasserBLIcK/BfG & Zuständige Behörden der Länder, 01.04.2021; status: last updated 1 April 2021; downloaded 19 July 2021 at 14:08). Background data source: Bayerisches Landesamt für Umwelt, www.lfu.bayern.de.**

subsequently averaged over 150 m depth intervals. The data were used to validate the normal compaction trends (NCT) determined by Drews et al. (2018) with regard to mudstone density data.



### 2.3 Mechanical compaction deduced from porosity-velocity relationships

Due to the mudstones' largely irreversible elastoplastic compaction behaviour, the degree of mechanical mudstone compaction provides a good first-order estimate of the maximum mean effective stress (Baig et al., 2019; Corcoran and Doré, 2005; Djéran-Maigre et al., 1998; Giles et al., 1998; Goulty, 1998; Hedberg, 1936; Hillis, 1995; Issler,

1992; Luo and Vasseur, 1995; Magara, 1980; Menpes and Hillis, 1995), hence the maximum burial depth, thereby assuming that the vertical stress represents the largest principal stress and vertical effective stress gradient, is known.

Mechanical compaction in terms of porosity decrease and velocity increase of both Mesozoic and Cenozoic mudstones from the North Alpine Foreland Basin have been previously investigated as a function of vertical

effective stress by Drews et al. (2018). The North Alpine Foreland basin is situated directly south of the study area (Figure 1 and Figure **3**) and uplift since maximum basin subsidence is estimated to have not exceeded more than ~500 m there (Baran et al., 2014; Drews et al., 2018; Kuhlemann and Kempf, 2002; Zweigel et al., 1998). Thus the depth-related increase in mudstone compaction in the North Alpine Foreland Basin (NAFB) is likely a good analogue for our study area. Drews et al. (2018) determined a mudstone compaction trend which utilizes porosity

decay as a function of vertical effective stress, based on the exponential compaction law of Athy (1930) (eq. 1):

$$\emptyset_{sh} = \emptyset_{0\_sh} * Exp(-VES/C). \tag{1}$$

Equation 1 is the porosity decay function of Athy (1930) modified for vertical effective stress (VES) according to Heppard et al. (1998), Rubey and Hubbert (1959), and Scott and Thomsen (1993). $\emptyset_{sh}$ is the mudstone porosity at a particular depth. Following Drews et al. (2018) the mudstone porosity at the surface $\emptyset_{0\_sh}$ was set to 0.4

(dimensionless) and the compaction coefficient C to 31 MPa$^{-1}$.

The porosity-velocity relationship proposed by Raiga-Clemenceau et al. (1986) can then be used to derive a velocity vs. vertical effective stress relationship:

$$Vp = Vp_{shm} * (1 - \emptyset_{sh})^{x} \tag{2}$$

Equation 2 is the mudstone porosity-velocity relationship of Raiga-Clemenceau et al. (1986) where Vp is the p-

wave velocity in mudstones. For the NAFB, Drews et al. (2018) set the matrix velocity of mudstones Vpshm to 5076 m/s and x to 2. Alternatively, Ø can be substituted by the water-saturated mudstone bulk density ρb_sat using the following relationship:

$$\rho_{b\_sat} = \rho_{t} * (1 - \emptyset) + \rho_{f} * \emptyset \tag{3}$$

Where $\rho_{t}$ is the true or skeletal density of the mudstone and $\rho_{f}$ is the density of the pore-filling fluid with 1.0 g/cm³

for water. The maximum burial depth TVD$_{max}$ can then be estimated from VES:

$$TVD_{max} = VES/VES_{grad} \tag{4}$$

with the vertical effective stress gradient VES$_{grad}$ typically varying between 10-16 MPa/km in hydrostatically pressured sedimentary basins, derived from a vertical stress gradient of 20-26 MPa/km and a hydrostatic pore pressure of 10 MPa/km (Bjørlykke, 2015). For the NAFB, Drews et al. (2018, 2020) determined a vertical effective

stress gradient of 13 MPa/km, which will also be used for depth calculations in this study.





### 2.3.1 Porosity and density

Dry bulk densities $\rho_{b\_dry}$ and porosities $\emptyset_{Hg}$ of 72 clay-/mudstone samples have been measured with a mercury intrusion porosimeter ("Poremaster 60" by Quantachrome) which analyzes pore diameters in the range of 0.0036 - 950 μm under pressures of up to 60000 psia. Prior to measurements, samples were dried at 65°C until no change in mass could be determined for 24 hours. Thereby, cracks may have formed during sample preparation and dehydration (Klaver et al., 2012). In turn this might result in the intrusion of mercury into these cracks at low pressures, but associated data excursions are rather obvious and were removed prior to further analysis as proposed by Klaver et al. (2015). True (skeletal) densities $\rho_t$ were determined for a subset of 34 samples by applying Helium pycnometry ("Accupyk II 1345" by Micromeritics), which enables analysis of even smaller pores (0.22 nm) than mercury (3.6 nm) (Hedenblad, 1997; Krus et al., 1997). For samples lacking direct $\rho_t$ measurements, the mean true density $\rho_{t\_mean}$ was used for further calculations. Using bulk density $\rho_{b\_dry}$ and true density $\rho_t$, respectively $\rho_{t\_mean}$ the (effective) porosity $\emptyset_{calc}$ was calculated:

$$\emptyset_{calc} = 1 - \frac{\rho_{b\_dry}}{\rho_{t\_mean}} \tag{5}$$

### 2.3.2 Velocity modeling based on density/porosity measurements

Applying the porosity-velocity relationship (c.f., eq. 2) proposed by Raiga-Clemenceau et al. (1986), respectively the velocity-density relationship by using density instead of porosity values (c.f., eq. 5) then allows for the calculation of mudstone velocities. Calculating mudstone velocities from $\emptyset_{calc}$ yields $Vp_{calc}$, while mudstone velocities based on measured $\emptyset_{Hg}$ values are labelled $Vp_{calc-Hg}$.

### 2.3.3 Mudstone velocity

*In situ* mudstone velocities Vp were derived from near surface (15-45 m TVD, see Table 1) seismic refraction data acquired in the course of this study (see locations in Figure 1), published borehole geophone measurements (Buness and Bram, 2001), and downhole sonic log readings (Welz, 1994).

## 2.4 Mudstone composition

### 2.4.1 Mineralogy

For XRD-based whole rock mineralogical classifications the dried mudstone samples were crushed and grinded with the McCrone XRD mill and analysed by a X-ray diffractometer D5000 (Siemens). A qualitative Rietveld analysis of the resulting signal was then done with the DIFFRAC.SUITE software EVA and thereafter, semi-quantitatively with the DIFFRAC.SUITE software TOPAS 4.2 (both by Bruker).

### 2.4.2 Grain size analysis

Full disaggregation of the solid samples was achieved by applying the "saturation-freeze-thaw" method of Yang and Aplin (1997). Particle size analysis by sedimentation was done by a SediGraph III Plus by Micromeritics. The grain size classes are differentiated according to the geotechnical grain size classification scheme for soils (Deutsches Institut für Normung, 1987), where the clay fraction comprises particles <2 μm, the silt fraction particles of 2-63 μm, and sand particles are >63 μm. The grain size classification scheme follows Potter et al. (1980).





**2.5     Vitrinite reflectance**

Random vitrinite reflectance in oil (VR) was determined for 11 selected samples (Table 1) using a magnification of 100× in non-polarized light at a wavelength of 546 nm (Taylor et al., 1998). Yttrium-Aluminium-Garnet (R=0.899%) and Gadolinium-Gallium-Garnet (R=1.699%) standards were used for calibration. As the vitrinite

maturation is mainly affected by temperature as well as by the duration of maximum burial (Nöth et al., 2001) and only to a minor degree by pressure (Hunt, 1979), these measurements are strongly dependent on the evolving heat flow and therefore the geothermal gradient within a sedimentary basin (Suggate, 1998). Vitrinite reflectance depth profiles therefore have to be set up for a specific region of interest. However, heat flow and resulting geothermal gradient may have changed over time, and there are variables like the respective organofacies or the individual

reaction kinetics which may influence the transformation and ordering processes of vitrinites (le Bayon et al., 2011). A VR-depth-trend was constructed, based on published vitrinite reflectance data (Gusterhuber et al., 2012) and partly unpublished data for Cretaceous mudstones in the northern part of the NAFB in Austria, where the samples' burial depths were known to allow calibration (Figure 3). From the correlation between the measured sample vitrinite reflectance and the VR-depth-trend, the burial depth of Franconian Alb clay-/mudstones was

inferred. As the Mesozoic burial history of the northern part of the Upper Austrian Molasse Basin (Nachtmann and Wagner, 1987) is rather similar to the Franconian Alb area (Peterek et al., 1997; Schröder, 1987), a comparison between our samples and the developed VR-depth-trend is considered as reasonable.

**3     Results and discussion**

**3.1     Mudstone composition**

All 41 clay-/mudstone samples were analyzed in terms of their grain size classification (Figure 4A) and their mineralogical composition (Figure 4B) to ensure that we base our study on a rather homogeneous sample set in terms of grain size and mineralogical composition.

*Grain size classification*

Most of the claystone pit samples contain <10% of grains >63 µm (sand fraction), 40-60% of grains in the range

2-63 µm (silt fraction), and 40-60% of grains <2 µm (clay fraction). Therefore the majority of samples classifies as "mudstones" or "claystones" (Figure 4A). Exceptionally high clay fraction percentages were observed for few samples from the claystone pit Großheirath as well as for core samples from Mistelgau and Zankschlag (Figure 4A). The fact that cores from one well location were sampled at various depth levels, explains the large spread in grain size classifications, particularly for the Zankschlag well samples, where several meters of cores were

analysed. Two Zankschlag core samples with increased sand and decreased clay contents (Figure 4A) were excluded from further analysis as they classify as sandy mudshales rather than "pure" mud- or clayshales in the classification scheme of Potter et al. (1980). This is because major deviations in petrophysical properties (e.g. porosity and p-wave velocity) of mudstones and compaction behaviour are reported for samples with increasing sand admixture and <40% clay content (Marion et al., 1992).

*Mineralogical composition*

Clay mineralogical studies of marine Jurassic clays and marls in our study area by Krumm (1965) have shown a dominance of illite and muscovite over kaolinite and low quantities of chlorite and vermiculite. Mineral compositions are hardly varying even over large distances and compositional variations are only observed among


different stratigraphic units. Clay mineralogy based mudstone compaction should therefore be relatively uniform

for the investigated mudstone samples and hence, comparable to each other. The mineralogical compositions of

analyzed clay- and mudstone samples are shown in Figure 4B. There is a very limited range of variation between

the individual claystone pit samples, most of which contain on average 44 wt.% clay minerals besides ~42 wt.%

accessory minerals (mainly quartz, pyrite, or rutile) and 14 wt.% carbonate minerals. In most samples, the amount

of carbonate minerals was low and in the range of 2-14 wt.%. Samples, that contained >40 wt.% of calcareous

minerals were excluded from further analysis. Increased calcite content in mudstones is often associated with early

cement stabilization, leading to increased strength (Horpibulsuk et al., 2010) that might counteract mudstone

compaction during burial.

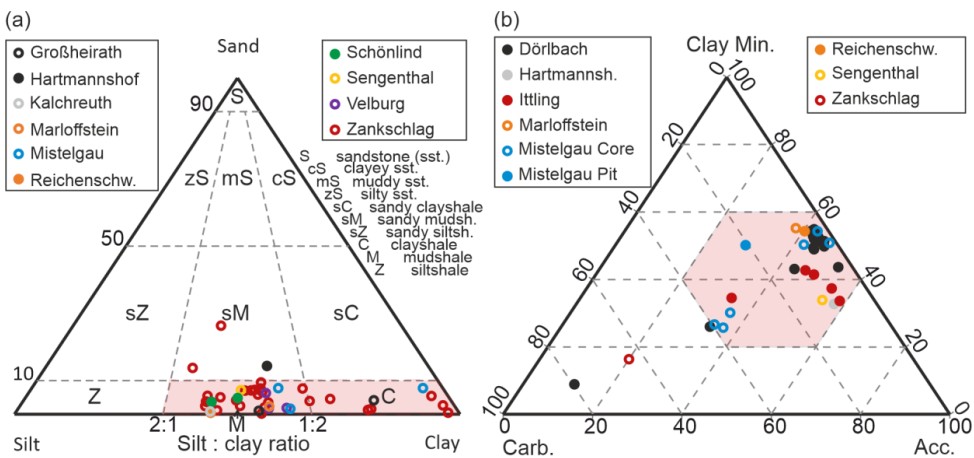

**Figure 4: a) Grain size classification of mudstone samples (according to Potter et al., 1980; plot layout modified from**
**Lindholm, 2012) with sand (>63 µm), silt (2-63 µm) and clay (<2 µm) fractions as end members of the ternary plot. Only**
**samples within the fields coloured in red were used for further measurements. b) Ternary plot of XRD-based mudstone**
**composition illustrating relative abundance of clay minerals (e.g. illite, smectite, kaolinite, chlorite, etc.), carbonate**
**minerals (e.g. calcite, dolomite, ankerite, siderite, etc.) and accessory minerals (Acc.) including quartz pyrite, rutile, etc..**
**Only samples within the reddish boxes were included in further analysis.**


## 3.2    Mudstone velocity data

Compressional p-wave velocities of Jurassic mudstones, which have been retrieved from shallow seismic

refraction surveys (see locations in Figure 1) and sonic log data of the shallow Zapfendorf borehole (Welz, 1994)

(Table 1) increase and converge towards velocities of 2000-3500 m/s at depths of 15 m below ground level (Figure

5A). We infer from this, that below a depth of 15 m, unloading related processes are negligible and therefore

selected only velocities from depth >15 m for further analysis.

Mudstone velocity vs. true vertical depth (TVD) plots for normally pressured Mesozoic and Cenozoic mudstones

in the NAFB (Drews et al., 2018; their Figure 4) show that mudstone compaction can be approximated by a single

trend with the calculated normal compaction trend (NCT) derived from the combination of a modified Athy

equation (c.f., eq. 1) and the porosity-velocity transform (c.f., eq. 2) of Raiga-Clemenceau et al. (1986). Drews et

al. (2018) also determined the systematic depth-dependent velocity increase of Mesozoic and Cenozoic mudstones

as a function of vertical effective stress (derived from in situ measured pressures from drill-stem, production and

wireline formation tests and associated mudstone velocities), well captured by the calculated NCT on a basin-wide

scale (Figure 5B).


Relating maximum mudstone velocities of 2500-3500 m/s, measured in Jurassic mudstones of the Franconian Alb
area (Figure 5A) to the NCT established by Drews et al. (2018) correlates them to vertical effective stresses in the
range of 10-25 MPa (Figure 5B) and would roughly equate to 700-2000 m true vertical depth according to the
NCT of (Drews et al. 2018; their Figure 4).

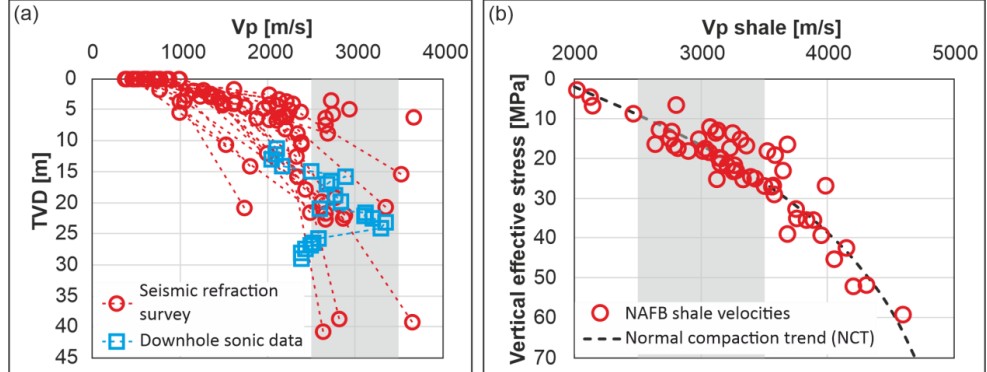

**Figure 5: (a) Clay-/mudstone velocities from field measurements in the Franconian Alb area versus true vertical depth**
**(TVD); data sources are shallow seismic refraction surveys (this study) and downhole sonic log data of the shallow**
**Zapfendorf well (Welz, 1994). (b) Mudstone velocities from sonic log and vertical seismic profile (VSP) data of deep**
**wells in the North Alpine Foreland Basin (NAFB) as a function of vertical effective stress (derived from drill stem and**
**production tests and wireline formation pressure tests) (redrawn from Drews et al., 2018). All data shown represent**
**hydrostatically pressured mudstone sections. The black dashed line represents the normal compaction trend (NCT)**
**determined by Drews et al. (2018). The grey background-boxes mark the maximum velocity range of clay-/mudstones**
**determined by field measurements in the Franconian Alb area.**

### 3.3    Integrating mudstone porosity and velocity data

Dry bulk densities $\rho_{b\_dry}$ and porosities $\varnothing_{Hg}$ were analyzed from 72 samples by Hg-intrusion porosimetry and true
(skeletal) densities $\rho_t$ with an average value $\rho_{t\_mean}$ of 2.74 ± 0.05 g/cm³ (Figure 6A) of 34 clay pit and shallow
drill core samples (Table 1) were determined by He-pycnometry. Mudstone porosities were also calculated ($\varnothing_{calc}$),
based on bulk densities $\rho_{b\_dry}$ and true (skeletal) densities $\rho_{t\_mean}$ (eq. 5). We preferred the calculated porosity values
rather than Hg-porosities because continued mercury intrusion even at the device's maximum injection pressure
(see inset in Figure 6B) suggested that micropores <0.003 μm were not fully involved in the measurement. The
cross-plot of calculated porosities $\varnothing_{calc}$ versus measured porosities $\varnothing_{Hg}$ reveals major discrepancies due to the
incomplete involvement of micropores by using Hg-porosities (Figure 6B). The relation between downhole
mudstone velocities and bulk densities is well captured by the NCT established by Drews et al. (2018) (Figure
7A). Figure 7B compares mudstone velocities $Vp_{calc}$ with $Vp_{calc-Hg}$. The values reveal a positive linear relationship,
but with significant diversions towards faster $Vp_{calc-Hg}$ values and a clustering of $Vp_{calc}$ values at 3000-3500 m/s
(Figure 7B).

As shown by the boxplot summary (Figure 8A), calculated mudstone velocities $Vp_{calc}$ applying $\varnothing_{calc}$ are
considerably lower (average 3300 m/s) than $Vp_{calc-Hg}$ applying $\varnothing_{Hg}$ (average 3900 m/s) due to the incomplete
involvement of micropores in $\varnothing_{Hg}$ based calculations (cf., Figure 6B and Figure 7B). Calculated mudstone


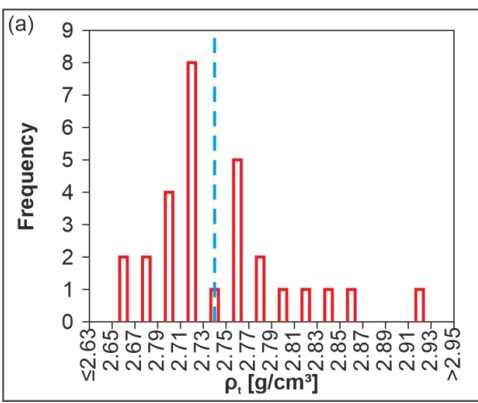
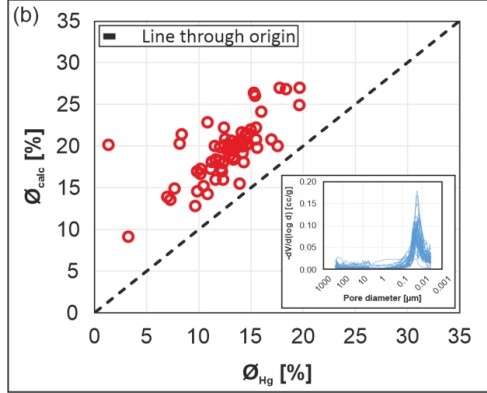

**Figure 6: a) Histogram of Helium (He) pycnometry derived true densities ρt of mudstones of the Franconian Alb, yielding an average value of 2.74 g/cm³ (vertical red dashed line); b) Hg-intrusion porosimetry derived porosities Ø$_{Hg}$ versus calculated porosities Ø$_{calc}$ based on the quotient of bulk densities ρb_dry and mean true densities ρt_mean. Inset indicates continued mercury intrusion even at the device's maximum injection pressure, suggesting that Hg-intrusion porosimetry does not include the entire micropore spectrum.**

velocities Vp$_{calc}$ are higher compared to *in situ* measured mudstone velocities derived from seismic refraction surveys (Vp$_{seis}$ average 2600 m/s) and shallow sonic log data (Vp$_{log}$ average 2800 m/s) from the Franconian Alb area. This is most likely method-related, as Vp$_{calc}$ values represent lab-based measurements on small, homogeneous sample volumes which are analyzed under controlled conditions, while *in situ* measured velocities

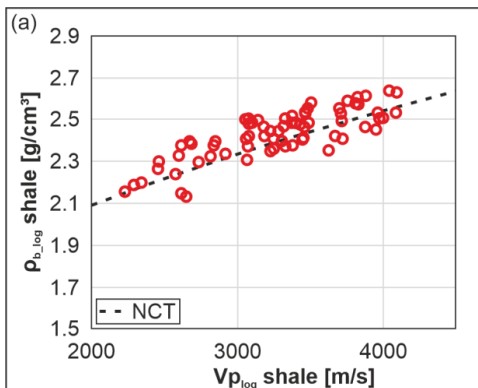
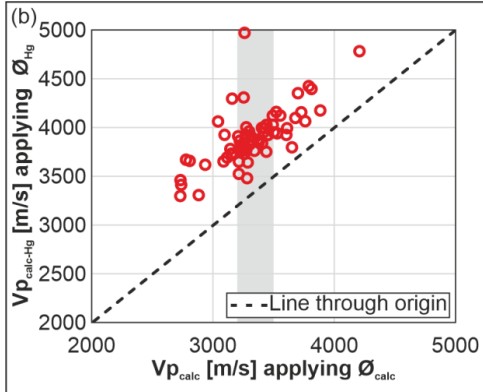

**Figure 7: Mudstone velocity-density model. a) P-wave velocity (Vplog) from sonic log and vertical seismic profile data as a function of bulk density log data ρb_log of deep wells in the NAFB (after Drews et al., 2018). The black dashed line represents the NCT of Drews et al. (2018). b) Calculated mudstone velocities Vp$_{calc}$ applying Ø$_{calc}$ vs. Vp$_{calc-Hg}$ using Ø$_{Hg}$. Grey bar highlights clustering of Vp$_{calc}$ values at 3000-3500 m/s.**

refer to larger volumes and hence, might be influenced by factors such as variations in grain size, compaction, pore water saturation, and discontinuities.

Referencing mudstone velocities to the mudstone velocity trend of Drews et al. (2018) derived from hydrostatically pressured mudstones in the NAFB (Figure 5B) views Vp values as a function of vertical effective stress (VES). Any uplift, although reported <500 m for the mudstones in the North Alpine Foreland Basin (Baran et al., 2014) could lead to an underestimation of our burial depth estimation by the respective amount, but will be neglected in our calculations as it is within the range of uncertainty. The majority of field velocity data from seismic refraction





survey $Vp_{seis}$ and shallow sonic log data $Vp_{log}$ (Welz, 1994) indicate a paleo-vertical effective stress in the range of 7-19 MPa (average 10 MPa for seismic refraction and 14 MPa for sonic log), while calculated velocities $Vp_{calc}$ and $Vp_{calc-Hg}$ yield higher values in the range of 19-25 MPa and 22-90 MPa, respectively (average 23 MPa) (Figure 8B). This could be due to the scale of the measurement: While the in situ field velocity data were measured roughly on a meter-scale and most likely also captured larger unloading structures due to the shallow present-day burial

depth, the measured porosity data are derived from cm-sized samples, which most likely are not as much affected by unloading.

Applying an average vertical effective stress gradient of 13 MPa/km to field velocity data of mudstones $Vp_{seis}$ and $Vp_{log}$ yields a maximum burial depth for Franconian Alb area samples of 0.0-1.8 km ($0.9 \pm 0.4$ km mean), whereas $Vp_{calc}$ and $Vp_{calc-Hg}$ yield 1.0-3.5 km ($1.8 \pm 0.4$ km mean) versus 1.7-6.9 km ($2.8 \pm 0.8$ km mean) burial, respectively

(Table A- 1). A lower stress gradient, associated with a less consolidated overlying rock column, would result in elevated maximum burial depths. In the unlikely case of a higher stress gradient, reflecting an overlying rock column of much denser lithology, this would yield decreased maximum burial depth values. Therefore, the applied VES gradient of 13 MPa/km and resulting maximum burial depth values represent a lower bound. Hence depth-corrected field velocity data and lab porosity data based on $Ø_{Hg}$ suggest that about 0.2-0.8 km (0.3 km mean)

respectively 1.1-6.3 km (2.2 km mean) of post-Jurassic sediments were removed in the Franconian Alb area since deposition (Figure 8C). Lab porosity data $Ø_{calc}$, however, are considered as more reliable, and suggest 0.9-1.4 km (1.1 km mean) of post-Jurassic overburden.

All these values must be corrected by their actual sample burial depth. However, instead of substracting individual corrections for the Upper Jurassic strata thickness at each sample location, an average value was removed. This is

related to the fact that only remnants of Upper Jurassic limestones are preserved with up to 200 m thickness, but an unknown amount of Upper Jurassic sediments was eroded in large parts of the Franconian Alb. Hence, their original paleo-thicknesses can only be inferred from seismic data in the NAFB, where Bachmann et al. (1987) determined a general value of 0.6 km for the thickness of the Upper Jurassic Malm unit. This thickness was thus removed from the calculated burial depth values.

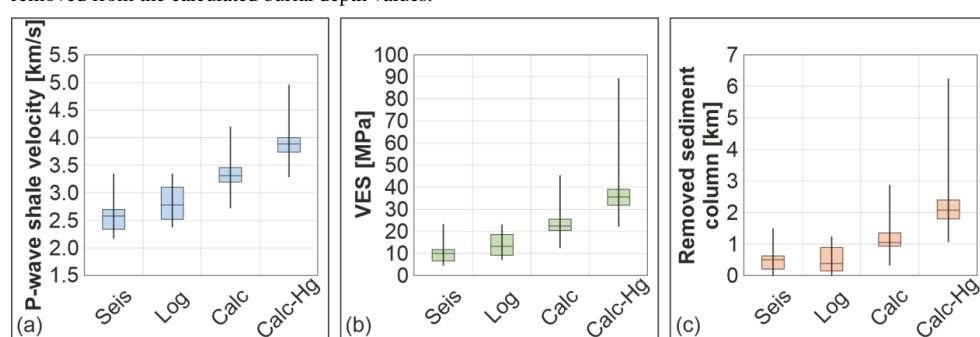


**Figure 8: Box plot summary of mudstone compaction results in the Franconian Alb area. a) Boxplot summary of measured and calculated mudstone velocity ranges from shallow seismic refraction data $Vp_{seis}$ (Seis), shallow sonic log data $Vp_{log}$ (Log) (Welz, 1994), and calculated velocity $Vp_{calc}$ applying $Ø_{calc}$ (Calc) and $Vp_{calc-Hg}$ applying measured $Ø_{Hg}$ values (Calc-Hg). b) Same as a), but velocities have been referenced to equivalent vertical effective stress (VES)**

**according to the normal mudstone compaction trend (NCT) of Drews et al. (2018) in the NAFB. c) Same as b), but showing thickness ranges of removed post-Jurassic sediment columns when applying an average vertical effective stress (VES) of 13 MPa/km. An average thickness of 0.6 km has been substracted for removed Upper Jurassic (Malm) sediments.**





Furthermore, no samples were corrected for their distances to the Middle Jurassic-Upper Jurassic interface at each
location. As the Upper Jurassic limestones are missing at most sample locations, so is the knowledge on the actual
distance to the Middle Jurassic-Upper Jurassic interface. Estimates for the former position of this interface in the
Franconian Alb area were only done by von Freyberg (1969). As the majority of the investigated samples are of
Middle Jurassic age, only interpolated values based on a georeferenced map of von Freyberg (1969) are available
for the sample locations. Because of the thicknesses of Middle Jurassic sediments of 20-170 m or even less (Meyer
and Schmidt-Kaler, 1996), we consider the neglect of these sediments to lie within the uncertainty range and did
not include them in the calculation of the removed sediment columns in Figure 8C. A summary of burial depth
and amount of removed sediment calculations at each sample location, based on a variety of different input
parameters is given in Table A- 1.

### 3.4    Vitrinite reflectance

Vitrinite reflectance values of Upper Triassic to Middle Jurassic mudstone samples from the Franconian Alb vary
between 0.32 %Ro and 0.61 %Ro with a mean of 0.49 %Ro and a correlation coefficient of R² = 0.76 with true
vertical depth (TVD) (Figure 9).

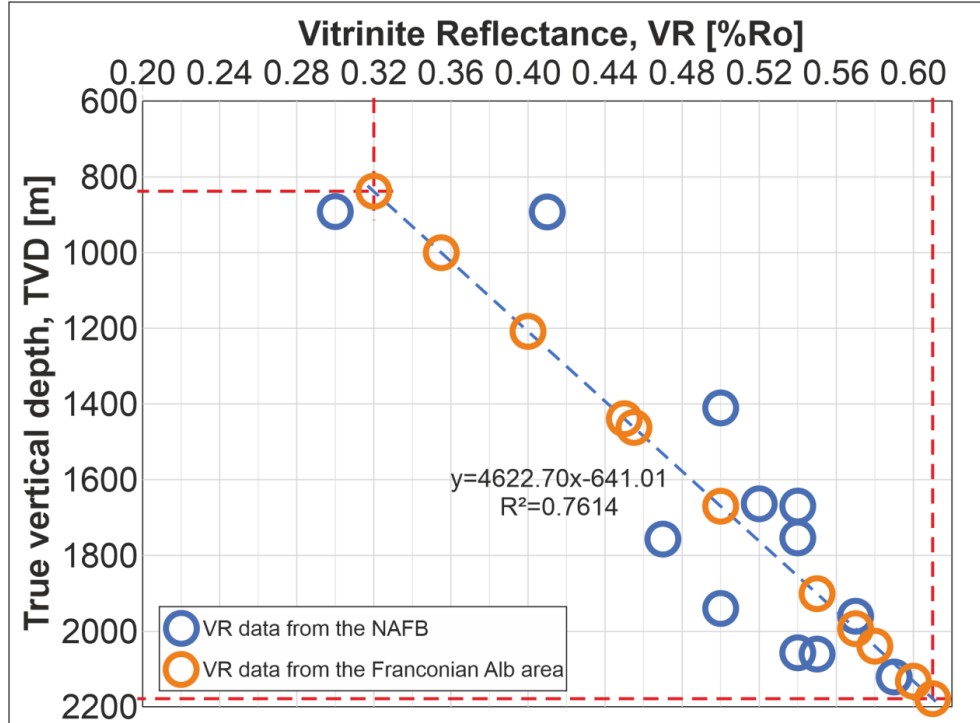

**Figure 9: Comparison of Franconian Alb area VR data (this study) to the VR-depth-trend (TVD) derived from
published (Sachsenhofer, 2001) and unpublished vitrinite reflectance data (Sachsenhofer, written comm. 2021) of the
northern, Austrian part of the NAFB. R² is the coefficient of determination. The range of vitrinite reflectance values of
our samples and inferred burial depths of c. 800-2200 m are indicated by the red dashed lines.**

As no information on the paleo heat flow in this region is available, no vitrinite reflectance evolution with depth
could be modelled for the study area. However, a comparable VR-depth-trend is derived from published





(Sachsenhofer, 2001) and unpublished (written comm. Prof. R. Sachsenhofer 2021) vitrinite reflectance data of
       Upper Triassic to Middle Jurassic mudstones from the northern part of the NAFB in Austria (Figure 3). Our results
       can be related to these, as they presumably have a similar thermal history. Samples from the Austrian part of the
       NAFB show vitrinite reflectances of 0.3-0.6 %Ro developed at sampling depths of ~900 - 2200 m (Figure 9).
       Applying this VR-depth trend to Franconian Alb VR data, reveals a similar paleo-burial depth range of 800-2200

m for the Franconian Alb area samples. Hence, applying VR data our Lower Jurassic Franconian Alb samples
       probably experienced a maximum burial depth average of ~1650 m and, considering ~600 m thickness for Upper
       Jurassic sediments, a removed post Jurassic sediment column of ~1050 m is calculated.

### 3.5     The Franconian Alb burial history in a regional context

       Our burial depth calculations for the Early to Middle Jurassic mudstones of the Franconian Alb area suggest a

burial depth of at least 900 m, based on downhole and shallow seismic refraction mudstone velocities, but rather
       ~1700 m as inferred from calculated porosities $\emptyset_{calc}$ and VR data as any unloading and drying effects can be ruled
       out in these data sets (Figure 10). A strong overestimation of maximum burial depths derived from $\emptyset_{Hg}$ porosity
       values is displayed in Figure 10C but has low reliability due to the incomplete micropore involvement (Figure
       6B). As the thicknesses of Early Jurassic strata (~20 m in the southern and ~100 m in the northern Franconian

Alb), of Middle Jurassic strata (~150 m: Meyer and Schmidt-Kaler, 1996) and of Late Jurassic sediments (~600
       m in the neighbouring NAFB: Bachmann et al., 1987) are roughly known, cumulative Jurassic sediment
       thicknesses are substracted from maximum burial depth to get values for removed post-Jurassic (Cretaceous plus
       Cenozoic) sediment thicknesses. The maximum overburden results for each location in the Franconian Alb area
       and each calculation method are listed in Table A- 1.

Our Vitrinite Reflectance data (Figure 10A and D), indicating burial depths of 0.8-2.2 km (mean 1.7 km), correlate
       very well with burial depth of ~1.7 km inferred from calculated porosities $\emptyset_{calc}$ applying He-pycnometry derived
       mean true densities $\rho_{t\_mean}$ and bulk densities $\rho_b$ (Figure 10B) (see Table A- 1).

       West of the Franconian Line, AFT-data (Hejl et al., 1997) and field mapping- and literature-based interpretations
       (sedimentological studies, thermochronological data, radiometric age data, etc.) suggest deposition and subsequent

removal of > 1000 m of Cretaceous and Cenozoic sediments (Peterek and Schröder, 2010; Schröder, 1987;
       Schröder et al., 1997) of which only c. 320 m of Upper Cretaceous strata are preserved (Dill, 1995). Hence,
       compared to the more distal western parts of the Franconian Alb, strongly increased depositional thicknesses along
       the front of the Franconian Line can be considered due to the uplift and major exhumation of the Bohemian Massif
       to the east, combined with westward thrusting, and syntectonic deposition of the eroded material (Meyer, 1996;

Peterek and Schröder, 2010; Walter, 2007).

       Results of the AFT- and (U-Th)/He-analysis of von Eynatten et al. (2021) on the other hand suggest burial depths
       of 3.0-4.0 km for exposed Triassic sedimentary rocks in large parts of central Germany, including the Franconian
       Alb. Applying these values, about 0.9 km of Jurassic and 2.1-3.1 km of Cretaceous/Cenozoic sediments would
       have been removed which exceeds our estimations for removed Cretaceous/Cenozoic sediments by ~1.1 km. This

discrepancy can be explained either by the fact that von Eynatten's Franconian Platform sample locations, c. 20
       km to the north of our study area, experienced a different subsidence/burial history or by the applied geothermal
       gradient which von Eynatten et al. (2021) estimated at only 30°C/km. This gradient contrasts to an elevated
       regional geothermal gradient of 38°C/km postulated by de Wall et al. (2019) in the vicinity of the Franconian Alb






**Figure 10: Areal distribution of calculated mean burial depth of sampled Lower/Middle Jurassic mudstones in the Franconian Alb area, based on two different methods: a) burial depths derived from the correlation between the NCT of Drews et al. (2018) and reliable in situ p-wave velocities, including shallow seismic refraction data (Vp_seis), shallow sonic log data (Vp_log, Welz, 1994), and borehole geophone data (Vp_geo, Buness and Bram, 2001). Furthermore, burial**

**depth calculations based on the correlation between the NAFB derived VR-depth-trend and Franconian Alb area VR**

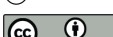

**data are included. b) Burial depths inferred from Vp$_{calc}$ based on porosities Ø$_{calc}$ c) Burial depths inferred from Vp$_{calc}$-Hg based on porosities Ø$_{Hg}$ . d) Detailed map of the blue dashed box in a). Fills of sampling points according to color scheme for total eroded thicknesses. See Table A- 1 for detailed results. Background data source: Bayerisches Landesamt für Umwelt, www.lfu.bayern.de.**

close to Mistelgau. Elevated geothermal gradients of >40°C/km are also observed in the area around Mürsbach (Bauer, 2000; Kämmlein et al., 2020). If the increased geothermal gradient applies also to the area investigated by von Eynatten et al. (2021), significantly lower burial depths would result in his calculations. However, the elevated geothermal anomaly is rather focussed to an area c. 20 km N of Bamberg (Figure 1) and quickly diminishes towards the south and east. (on Eynatten et al. (2021) also state that the magnitudes of exhumation and erosion are

remarkably reduced towards the eastern Franconian Alb margin. We, therefore, think that our estimates of removed post-Jurassic sediments for the Franconian Alb area are more realistic and do not contradict but rather support and complement the results of von Eynatten et al. (2021). Bachmann et al. (2002) argue that no Cretaceous sediments were deposited in the western part of the Franconian Alb area. This conclusion can most likely be related to the more distal-to-source position of their study area, positioned between Tübingen and Würzburg, compared to ours

(Franconian Alb area). As Cretaceous sediments in the Franconian Alb area were most likely sourced from the Bohemian Massif towards the east (Voigt et al., 2008), a reduced sediment supply to positions more distal to the source can be expected. Westward decreasing Cretaceous sediment columns, as proposed by Meyer (1996) and Peterek and Schröder (2010), support this interpretation.

### 3.6      Spatial distribution of post-Jurassic sediment overburden

The lateral variation of calculated burial depths derived from two independent data sets (Figure 10A-D) is showing no regional trends nor are areas of increased or reduced burial depth noticable. Only in the case of the porosity-derived burial depth estimations (Figure 10B), a trend towards increased amounts of post Lower Jurassic paleo-thicknesses in the northwestern part of the Franconian Alb can be conjectured, though this impression is based on a sparse data density in the area of interest.

Additional information comes from published AFT- and measured VR-data. From the VR results, no distinct differential vertical movements between various parts of the Franconian Alb can be inferred. According to von Eynatten et al. (2021), however, AFT- and (U-TH)/He-data indicate that Triassic sediments were less deeply buried next to the Bohemian Massif boundary in the east (<<3-4 km) compared to the central part of their study area (3-4 km), situated to the north of the Franconian Alb area. The discrepancy to our results (~1 km) can be explained

with the doming model of von Eynatten et al. (2021), as their analyzed Franconian Platform sample set was taken closer to the doming centre which is located further to the north of our study area. Hence, our study area was most likely less affected by doming-related processes. The AFT-results of Hejl et al. (1997) and the sedimentological observations of Schröder (1987) and Peterek and Schröder (2010) additionally suggest that higher sediment thicknesses (~2 km) were deposited directly west of the Franconian Line compared to the more distal-to-source

parts. The more distal-to-source locations of the majority of our samples most likely explains these reduced burial depths. Reasons for reduced sediment removal in the southwestern part of the study area are given by Peterek and Schröder (2010). They suggest temporarily reduced erosion rates in this area due to the coverage by Neogene lake sediments that protected underlying Mesozoic sediments from erosion.

In summary our data suggest that considerable amounts of post-Jurassic sediments must have been removed from

the investigated area. Having information on the paleo-stress conditions during burial of nowadays surface-exposed sedimentary rocks is a key for relating their petrophysical properties to their deeply buried analogues. Our



results indicate that the Upper Jurassic "Malm" carbonates, which are exposed in the Franconian Alb area and plunge southwards to depths of up to 5500 m in the Alpine foreland (Bachmann et al., 1987), constitute suitable analogues for reservoirs drilled at equivalent burial depths of ~1050 m in the NAFB. This would directly apply to the geothermally productive Malm reservoirs in the proximal north of Munich and in the Moosburg-Landshut area (Figure 3).

## 4 Conclusion

This study aimed to quantify eroded thicknesses of post-Jurassic sediments that were originally deposited in the Franconian Alb area, forming the south-eastern part of the German Basin. We thereby took advantage of the presence of widely distributed Lower Jurassic mudstones and their inelastic compaction behaviour, well recording maximum burial depth by their petrophysical properties. From various locations distributed over the Franconian Alb a large number of mudstone density and porosity measurements were performed and complemented by vitrinite reflectance and both new and published in-situ p-wave velocity data from seismic surveys and and downhole logging. These datasets were subsequently related to a compaction-depth-trend that was calibrated on mudstones of the same stratigraphic unit in the NAFB to the south of our study area. From the velocity data, we conclude that the Lower/Middle Jurassic mudstones experienced a maximum overburden of ~900 m, of which ~600 m relate to Upper Jurassic and ~300 m to post-Jurassic sediments. More likely, however, are mean values of about 1100 m (total range 900 - 1400 m) of eroded Cretaceous/Cenozoic sediment thicknesses deduced from lab-based porosity and bulk density measurements as these rock parameters are less influenced by alteration, unloading effects and variable water saturation of *in situ* measured samples. Vitrinite reflectance data essentially confirm burial depths of ~1050 m post-Jurassic overburden (~1650 m for Lower/Middle Jurassic mudstones) derived from lab-based porosity and bulk density measurements. No clear trends for a lateral variance in reconstructed post-Jurassic sediment thicknesses were observed, although porosity- and bulk density-derived maximum burial depths suggest a slight thickness increase towards the northwest.

The results of this study provide a contribution to the post-Jurassic burial history of the Fanconian Alb region. We also realized that maximum burial calculations, based solely on refraction velocity measurements of near-surface mudstone samples may be heavily disturbed by relaxation and dehydration and thus would provide no reliable basis to set up normal-compaction-trends and maximum burial depth estimates. The integrated analysis of porosity-, bulk density-, p-wave velocity, and VR measurements that are related to calibrated depth-trends, however provide rather uniform estimates for the maximum amount of sediment overburden in concert with other studies. Quantifications of eroded sediment thicknesses and maximum overburden in turn will help to improve the understanding of Upper Jurassic diagenetic conditions and reservoir properties. In terms of equivalent maximum burial depths, Franconian Alb Malm strata can be considered as ideal outcrop analogues for Malm thermal water aquifers in the Munich-Moosburg-Landshut area in the NAFB.

## 5 Acknowledgements

This study has been funded by the Bavarian State Ministry of Science and the Arts through the frame work of the Geothermal-Alliance Bavaria. We are grateful that the "Autobahndirektion Nordbayern" (Sibylle Glück) provided us with cores from various locations and also to the TUM Hydrogeology group headed by Prof. Einsiedl and to



the head of the geothermal Energy group led by Dr. Zosseder, in particular to Daniel Bohnsack for providing us
access to their Helium pycnometer. We would also like to thank Prof. Reinhard Sachsenhofer and Dr. David Misch
for valuable input on the regional geology of the North Alpine Foreland Basin and existing thermal maturity data.
The work of David Misch was supported by the Austrian Science Fund FWF (grant no.: P 33883-N). For
conducting the grain size distribution measurements, we thank Dr. Ute Schmidt from the Institute of Geography
at FAU. Many thanks are also directed to Mr. Heinz Meyer (mayor of Burgthann) and Mr. Karl Lappe (mayor of
Mistelgau) who supported the realisation of this study by authorizing the drilling on their estates.

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


## 7    Appendix

**Table A- 1: Calculated mean burial depth results (incl. standard deviation) derived from the correlation between the normal compaction trend (NCT) after Drews et al. (2018) and calculated (Vp$_{calc}$ derived from Ø$_{calc}$ and Vp$_{calc-Hg}$ derived from Ø$_{Hg}$) as well as measured P-wave velocities Vp (from borehole geophone measurements Vp$_{geo}$ for B05 & B10 (Buness and Bram, 2001), sonic log data Vp$_{log}$ for Zd (Welz, 1994), and seismic refraction survey Vp$_{seis}$). Additionally, the burial depth results from the correlation between the VR-depth-trend and measured VR are listed. From these**

**results, also the amount of removed post-Jurassic (post-Jur) sediments was estimated. In case of buried samples where the mean burial depth is not equal to the total amount of eroded sediments, the amount of total sediment removal was additionally calculated. Values smaller than zero are excluded as they indicate unrealistically low burial depths, meaning that these samples were deposited later than the Middle Jurassic, although they are pre-Upper Jurassic sediments. Location abbreviations and associated locations and sampled stratigraphic units are listed in Table 1 and**

**illustrated in Figure 1.**





| Location | Calculated depths (m) | Method used for depth calculation | | | |
|---|---|---|---|---|---|
| | | $Vp_{calc}$ | $Vp_{calc-Hg}$ | $Vp$ | $VR$ |
| B05 | Mean sample burial depth | - | - | 1145 | - |
| | Total post-Jur thickness | - | - | 218 | - |
| B10 | Mean sample burial depth | - | - | 856 | - |
| | Total post-Jur thickness | - | - | 401 | - |
| Db | Mean sample burial depth | 1553 ± 363 | 2861± 310 | - | 1971 ± 70 |
| | Total sediment removal | 1544 ± 363 | 2851 ± 306 | - | 1957 ± 60 |
| | Total post-Jur thickness | 944 ± 363 | 2251 ± 306 | - | 1357 ± 60 |
| Gh | Mean sample burial depth | 2448 ± 83 | 4038 ± 119 | - | 1994 |
| | Total post-Jur thickness | 1847 ± 83 | 3438 ± 119 | - | 1393 |
| Ha | Mean sample burial depth | 1649 | 2704 | - | 1451 ± 12 |
| | Total post-Jur thickness | 1049 | 2103 | - | 850 ± 12 |
| Itt | Mean sample burial depth | 2068 ± 151 | 3207 ± 117 | - | - |
| | Total sediment removal | 2047 ± 151 | 3186 ± 117 | - | - |
| | Total post-Jur thickness | 1447 ± 151 | 2586 ± 117 | - | - |
| Kr | Mean sample burial depth | 1706 | 2677 | - | 2133 |
| | Total post-Jur thickness | 1106 | 2077 | - | 1533 |
| Ms | Mean sample burial depth | 949 | 1940 | - | - |
| | Total post-Jur thickness | 349 | 1339 | - | - |
| Mg Core | Mean sample burial depth | 1851 ± 157 | 2837 ± 132 | - | 1023 ± 185 |
| | Total sediment removal | 1845 ± 156 | 2831 ± 131 | - | 1019 ± 183 |
| | Total post-Jur thickness | 1245 ± 156 | 2231 ± 131 | - | 419 ± 183 |
| Mg Pit | Mean sample burial depth | 959 | 1861 | - | 1000 |
| | Total post-Jur thickness | 358 | 1261 | - | 400 |
| Rs | Mean sample burial depth | 1930 | 2801 | - | - |
| | Total post-Jur thickness | 1325 | 2196 | - | - |
| | Mean sample burial depth | - | - | 793 ± 372 | - |





| | | | | | |
|---|---|---|---|---|---|
| Seismic refraction survey | Total sediment removal | - | - | 765 ± 380 | - |
| | Total post-Jur thickness | - | - | - | - |
| Sl | Mean sample burial depth | 1675 ± 15 | 2920 ± 55 | - | 2179 |
| | Total post-Jur thickness | 1075 ± 15 | 2320 ± 55 | - | 1579 |
| St | Mean sample burial depth | 1305 ± 98 | 2228 ± 31 | - | 2040 |
| | Total post-Jur thickness | 704 ± 98 | 1627 ± 31 | - | 1440 |
| Us | Mean sample burial depth | 2383 ± 183 | 3513 ± 558 | - | 1670 |
| | Total post-Jur thickness | 1783 ± 183 | 2913 ± 558 | - | 1070 |
| Vb | Mean sample burial depth | 1461 ± 241 | 2486 ± 590 | - | - |
| | Total sediment removal | 1420 ± 237 | 2445 ± 584 | - | - |
| | Total post-Jur thickness | 820 ± 237 | 1845 ± 584 | - | - |
| Zg | Mean sample burial depth | 1930 ± 437 | 2948 ± 1057 | - | - |
| | Total sediment removal | 1875 ± 438 | 2894 ± 1057 | - | - |
| | Total post-Jur thickness | 1275 ± 438 | 2294 ± 1057 | - | - |
| Zd | Mean sample burial depth | - | - | 1044 ± 394 | - |
| | Total sediment removal | - | - | 1021 ± 395 | - |
| | Total post-Jur thickness | - | - | 261 ± 395 | - |
| All | Mean sample burial depth | 1768 ± 441 | 2839 ± 812 | 931 ± 393 | 1659 ± 443 |
| | Total post-Jur thickness | 1135 ± 439 | 2206 ± 812 | 162 ± 416 | 1056 ± 442 |





852
853
854

Table A-2: Supplementary table of all measurement results. Italic Numbers represent values that are calculated from measurement results. Abbreviations: TVD = true vertical depth; GSF = grain size fraction; VR = Vitrinite reflectance; XRD = X-ray diffraction; Acc. min. = accessory minerals; N. s. a. = Northern study area; Cs. P. = Claystone Pit; B. g. = Borehole geophone; S. s. = Seismic survey

| Sample ID | Location | Sample type | Coordinates X | Coordinates Y | TVD (m) | $\rho_{b,dry}$ (g/cm³) | $\rho_t$ (g/cm³) | $\Phi_{Hg}$ (%) | $\Phi_{calc}$ (%) | $V_p$ (m/s) | $V_{p\,calc-Hg}$ (m/s) | $V_{p\,calc}$ (m/s) | GSF <2 µm (%) | GSF 2-63 µm (%) | GSF <63 µm (%) | VR %Ro | XRD Clay Minerals wt.-% | XRD CaCO₃ wt.-% | XRD Acc. min. wt.-% |
|---|---|---|---|---|---|---|---|---|---|---|---|---|---|---|---|---|---|---|---|
| US1-1b | Unterstürmig | Cs. P. | 4431270 | 5517510 | 0.50 | 2.37 | - | 7.25 | 13.63 | - | 4389 | 3806 | - | - | - | 0.50 | - | - | - |
| US1-2c | Unterstürmig | Cs. P. | 4431270 | 5517510 | 0.50 | 2.31 | - | 11.58 | 15.89 | - | 3989 | 3609 | - | - | - | - | - | - | - |
| ST1-T | Sengenthal | Cs. P. | 4461882 | 5455676 | 0.50 | 2.08 | - | 15.92 | 24.11 | - | 3607 | 2938 | - | - | - | - | - | - | - |
| ST2-T | Sengenthal | Cs. P. | 4461882 | 5455676 | 0.50 | 2.13 | 2.76 | 15.51 | 22.20 | - | 3642 | 3088 | 10.47 | 40.35 | 49.17 | - | 33.8 | 11.5 | 54.2 |
| RS1 | Reichenschwand | Cs. P. | 4455450 | 5486790 | 0.50 | 2.25 | 2.69 | 12.35 | 17.80 | - | 3920 | 3447 | 2.53 | 50.61 | 46.85 | - | 53.8 | 5.3 | 39.9 |
| GHI | Großheirath | Cs. P. | 4425662 | 5560960 | 0.50 | 2.34 | - | 7.73 | 14.84 | - | 4344 | 3701 | 2.61 | 30.38 | 67.01 | 0.57 | - | - | - |
| GH2-1 | Großheirath | Cs. P. | 4425662 | 5560960 | 0.50 | 2.36 | 2.76 | 6.99 | 13.84 | - | 4413 | 3788 | 5.72 | 30.90 | 63.37 | - | - | - | - |
| MGI | Mistelgau | Cs. P. | 4461603 | 5529789 | 0.50 | 2.01 | - | 18.32 | 26.75 | - | 3404 | 2737 | 7.87 | 19.85 | 72.28 | 0.36 | 49.9 | 20.4 | 28.7 |
| SL-H1 | Schönlind | Cs. P. | 4483436 | 5493298 | 0.50 | 2.20 | 2.85 | 12.02 | 19.69 | - | 3949 | 3291 | 2.28 | 53.88 | 43.84 | 0.61 | - | - | - |
| SL-S2 | Schönlind | Cs. P. | 4483436 | 5493298 | 0.50 | 2.20 | - | 11.48 | 19.93 | - | 3998 | 3271 | 4.20 | 49.69 | 46.11 | - | - | - | - |
| HH1-T | Hartmannshof | Cs. P. | 4468009 | 5485046 | 0.50 | 2.19 | 2.75 | 12.87 | 20.03 | - | 3873 | 3263 | 8.86 | 39.15 | 52.00 | - | 32.8 | 37.5 | 57.7 |
| HH1-K | Hartmannshof | Cs. P. | 4468009 | 5485046 | 0.50 | - | - | - | - | - | - | - | - | - | - | 0.45 | - | - | - |
| HH2-K | Hartmannshof | Cs. P. | 4468009 | 5485046 | 0.50 | - | - | - | - | - | - | - | - | - | - | 0.46 | - | - | - |
| MS | Marloffstein | Cs. P. | 4432488 | 5498710 | 0.50 | 2.01 | 2.67 | 17.73 | 26.86 | - | 3454 | 2729 | 1.41 | 47.31 | 51.28 | - | 54.9 | 6.7 | 37.6 |
| Kr | Kalchreuth | Cs. P. | 4437792 | 5489800 | 0.50 | 2.21 | 2.69 | 13.01 | 19.56 | - | 3861 | 3302 | 0.95 | 49.16 | 49.89 | 0.60 | - | - | - |
| ZS26_1 | Zankschlag | Core | 4463112 | 5476029 | 49.65 | 2.34 | - | 9.86 | 14.57 | - | 4145 | 3724 | 3.98 | 31.74 | 64.28 | - | - | - | - |
| ZS26_2 | Zankschlag | Core | 4463112 | 5476029 | 50.00 | - | - | - | - | - | - | - | 26.32 | 40.44 | 33.24 | - | 16.1 | 63.3 | 20.2 |
| ZS26_3 | Zankschlag | Core | 4463112 | 5476029 | 50.40 | 2.49 | - | 3.26 | 9.18 | - | 4775 | 4208 | 5.81 | 42.33 | 51.87 | - | - | - | - |
| ZS26_4 | Zankschlag | Core | 4463112 | 5476029 | 50.95 | 2.20 | - | 14.22 | 19.94 | - | 3754 | 3270 | 7.08 | 42.39 | 50.53 | - | - | - | - |
| ZS26_5 | Zankschlag | Core | 4463112 | 5476029 | 51.30 | 2.19 | - | 14.02 | 20.11 | - | 3772 | 3256 | 7.68 | 40.81 | 51.52 | - | - | - | - |
| ZS26_6 | Zankschlag | Core | 4463112 | 5476029 | 51.70 | 2.20 | - | 13.49 | 19.60 | - | 3818 | 3298 | 6.78 | 41.14 | 52.08 | - | - | - | - |





| Sample ID | Location | Sample type | Coordinates | | TVD | $\rho_{b,dry}$ | $\rho_t$ | $\Phi_{Hg}$ | $\Phi_{calc}$ | Vp | $Vp_{calc-Hg}$ | $Vp_{calc}$ | GSF | | | VR | XRD | | |
|---|---|---|---|---|---|---|---|---|---|---|---|---|---|---|---|---|---|---|---|
| | | | X | Y | m | g/cm³ | g/cm³ | % | % | m/s | m/s | m/s | <2 μm | 2-63 μm | >63 μm | %Ro | Clay Minerals | CaCO₃ | Acc. min. |
| | | | | | | | | | | | | | % | % | % | %Ro | wt.-% | wt.-% | wt.-% |
| ZS26_7 | Zankschlag | Core | 4463112 | 5476029 | 52.10 | 2.19 | 2.69 | 14.57 | 20.09 | - | 3724 | 3258 | 7.18 | 43.22 | 49.60 | - | - | - | - |
| ZS26_8 | Zankschlag | Core | 4463112 | 5476029 | 52.60 | 2.24 | - | 12.02 | 18.18 | - | 3949 | 3415 | 2.45 | 55.92 | 41.63 | - | - | - | - |
| ZS26_9 | Zankschlag | Core | 4463112 | 5476029 | 52.80 | 2.28 | 2.75 | 12.19 | 16.90 | - | 3934 | 3523 | 4.03 | 55.02 | 40.95 | - | - | - | - |
| ZS26_10 | Zankschlag | Core | 4463112 | 5476029 | 53.00 | 2.24 | - | 11.64 | 18.33 | - | 3983 | 3403 | - | - | - | - | - | - | - |
| ZS26_11 | Zankschlag | Core | 4463112 | 5476029 | 53.35 | 2.21 | 2.73 | 12.59 | 19.42 | - | 3898 | 3313 | 2.42 | 48.21 | 49.37 | - | - | - | - |
| ZS26_12 | Zankschlag | Core | 4463112 | 5476029 | 53.75 | 2.35 | - | 10.80 | 14.15 | - | 4059 | 3761 | 13.83 | 53.12 | 33.05 | - | - | - | - |
| ZS26_13 | Zankschlag | Core | 4463112 | 5476029 | 53.95 | 2.18 | - | 13.71 | 20.40 | - | 3799 | 3233 | 5.28 | 3.52 | 91.20 | - | - | - | - |
| ZS26_14 | Zankschlag | Core | 4463112 | 5476029 | 54.15 | 2.21 | 2.72 | 13.17 | 19.43 | - | 3847 | 3312 | 0.50 | 44.24 | 55.26 | - | - | - | - |
| ZS26_15 | Zankschlag | Core | 4463112 | 5476029 | 54.55 | 2.39 | - | 9.62 | 12.76 | - | 4168 | 3883 | 9.43 | 39.88 | 50.70 | - | - | - | - |
| ZS26_16 | Zankschlag | Core | 4463112 | 5476029 | 55.05 | 2.24 | - | 13.35 | 18.33 | - | 3831 | 3403 | 3.90 | 55.01 | 41.10 | - | - | - | - |
| ZS26_17 | Zankschlag | Core | 4463112 | 5476029 | 55.44 | 2.30 | 2.85 | 12.33 | 15.96 | - | 3922 | 3603 | 4.20 | 48.18 | 47.62 | - | - | - | - |
| ZS26_18 | Zankschlag | Core | 4463112 | 5476029 | 55.80 | 2.22 | - | 14.23 | 19.06 | - | 3754 | 3343 | 7.57 | 31.68 | 60.75 | - | - | - | - |
| ZS26_19 | Zankschlag | Core | 4463112 | 5476029 | 56.96 | 2.18 | - | 13.06 | 20.60 | - | 3856 | 3216 | 0.70 | 2.20 | 97.10 | - | - | - | - |
| ZS26_20 | Zankschlag | Core | 4463112 | 5476029 | 57.06 | 2.19 | - | 1.40 | 20.07 | - | 4960 | 3260 | 2.59 | 2.27 | 95.13 | - | - | - | - |
| ZS26_22 | Zankschlag | Core | 4463112 | 5476029 | 57.36 | 2.20 | 2.77 | 17.54 | 19.84 | - | 3469 | 3279 | 7.14 | 44.49 | 48.37 | - | - | - | - |
| ZS26_23 | Zankschlag | Core | 4463112 | 5476029 | 57.90 | 2.23 | 2.71 | 13.26 | 18.70 | - | 3838 | 3372 | 2.15 | 42.62 | 55.23 | - | - | - | - |
| ZS26_24 | Zankschlag | Core | 4463112 | 5476029 | 58.16 | 2.20 | - | 14.10 | 19.89 | - | 3765 | 3274 | 4.71 | 26.61 | 68.67 | - | - | - | - |
| ZS26_25 | Zankschlag | Core | 4463112 | 5476029 | 58.60 | 2.18 | 2.73 | 15.49 | 20.67 | - | 3644 | 3211 | - | - | - | - | - | - | - |
| ZS26_26 | Zankschlag | Core | 4463112 | 5476029 | 58.98 | 2.18 | - | 14.05 | 20.33 | - | 3769 | 3238 | - | - | - | - | - | - | - |
| ZS26_27 | Zankschlag | Core | 4463112 | 5476029 | 58.14 | - | - | - | - | - | - | - | 3.48 | 41.16 | 55.35 | - | - | - | - |
| ZS26_29 | Zankschlag | Core | 4463112 | 5476029 | 56.85 | - | - | - | - | - | - | - | 5.41 | 54.01 | 40.58 | - | - | - | - |
| ZS26_30 | Zankschlag | Core | 4463112 | 5476029 | 56.02 | - | - | - | - | - | - | - | 1.50 | 19.15 | 79.34 | - | - | - | - |
| ZS26_33 | Zankschlag | Core | 4463112 | 5476029 | 58.03 | 2.25 | - | 14.28 | 17.95 | - | 3749 | 3434 | 1.06 | 53.97 | 44.96 | - | - | - | - |




| Sample ID | Location | Sample type | X | Y | TVD (m) | $\rho_{b,dry}$ (g/cm³) | $\rho_t$ (g/cm³) | $\Phi_{Hg}$ (%) | $\Phi_{calc}$ (%) | Vp (m/s) | $Vp_{calc-Hg}$ (m/s) | $Vp_{calc}$ (m/s) | GSF <2 µm (%) | GSF 2-63 µm (%) | GSF >63 µm (%) | VR %Ro | Clay Minerals (wt.-%) | CaCO₃ (wt.-%) | Acc. min. (wt.-%) |
|---|---|---|---|---|---|---|---|---|---|---|---|---|---|---|---|---|---|---|---|
| ZS26_34 | Zankschlag | Core | 4463112 | 5476029 | 61.20 | 2.32 | - | 13.85 | 15.39 | - | 3787 | 3652 | - | - | - | - | - | - | - |
| V24_2 | Velburg | Core | 4475168 | 5452094 | 24.77 | 2.17 | - | 16.98 | 20.70 | - | 3517 | 3208 | - | - | - | - | - | - | - |
| V24_3 | Velburg | Core | 4475168 | 5452094 | 46.05 | 2.16 | 2.72 | 14.68 | 21.28 | - | 3714 | 3162 | - | - | - | - | - | - | - |
| V24_4 | Velburg | Core | 4475168 | 5452094 | 46.13 | 2.15 | - | 14.06 | 21.50 | - | 3769 | 3144 | - | - | - | - | - | - | - |
| V24_5 | Velburg | Core | 4475168 | 5452094 | 46.17 | - | - | - | - | - | - | - | 1.93 | 37.92 | 60.15 | - | - | - | - |
| V24_6 | Velburg | Core | 4475168 | 5452094 | 46.40 | 2.19 | 2.76 | 8.14 | 20.19 | - | 4305 | 3250 | - | - | - | - | - | - | - |
| V24_7 | Velburg | Core | 4475168 | 5452094 | 47.54 | 2.16 | - | 8.38 | 21.39 | - | 4283 | 3153 | - | - | - | - | - | - | - |
| V24_8 | Velburg | Core | 4475168 | 5452094 | 48.65 | 2.15 | - | 14.52 | 21.41 | - | 3728 | 3151 | - | - | - | - | - | - | - |
| V24_9 | Velburg | Core | 4475168 | 5452094 | 47.95 | 2.14 | 2.72 | 14.98 | 21.84 | - | 3688 | 3117 | - | - | - | - | - | - | - |
| V24_10 | Velburg | Core | 4475168 | 5452094 | 48.60 | 2.23 | - | 12.92 | 18.75 | - | 3869 | 3369 | - | - | - | - | - | - | - |
| V24_11 | Velburg | Core | 4475168 | 5452094 | 48.71 | - | - | - | - | - | - | - | 1.71 | 42.06 | 56.23 | - | - | - | - |
| V24_12 | Velburg | Core | 4475168 | 5452094 | 48.02 | 2.03 | - | 15.38 | 25.88 | - | 3653 | 2803 | 6.28 | 40.54 | 53.18 | - | - | - | - |
| V24_13 | Velburg | Core | 4475168 | 5452094 | 49.45 | 2.20 | - | 15.61 | 19.79 | - | 3634 | 3283 | - | - | - | - | - | - | - |
| V24_14 | Velburg | Core | 4475168 | 5452094 | 49.95 | 2.19 | - | 13.85 | 20.04 | - | 3787 | 3262 | - | - | - | - | - | - | - |
| V24_15 | Velburg | Core | 4475168 | 5452094 | 25.50 | 2.18 | - | 14.28 | 20.47 | - | 3749 | 3227 | - | - | - | - | - | - | - |
| V24_16 | Velburg | Core | 4475168 | 5452094 | 25.95 | 2.06 | - | 19.57 | 24.88 | - | 3301 | 2879 | - | - | - | - | - | - | - |
| V24_17 | Velburg | Core | 4475168 | 5452094 | 24.95 | 2.01 | - | 19.63 | 26.86 | - | 3296 | 2729 | - | - | - | - | - | - | - |
| MG_3.00-3.10 | Mistelgau | Core | 4461603 | 5529789 | 3.04 | - | - | - | - | - | - | - | - | - | - | - | 26.8 | 39.2 | 33.9 |
| MG_3.35-3.45 | Mistelgau | Core | 4461603 | 5529789 | 3.36 | - | - | - | - | - | - | - | - | - | - | - | 25.8 | 37.7 | 36.3 |
| MG_3.47-3.55 | Mistelgau | Core | 4461603 | 5529789 | 3.48 | - | - | - | - | - | - | - | - | - | - | - | 30.1 | 34.1 | 35.8 |
| MG_5.10-5.25 | Mistelgau | Core | 4461603 | 5529789 | 5.18 | 2.20 | 2.71 | 12.41 | 19.91 | - | 3914 | 3273 | - | - | - | 0.32 | 50.5 | 1.5 | 47.4 |
| MG_5.25-5.60 | Mistelgau | Core | 4461603 | 5529789 | 5.42 | 2.21 | 2.91 | 13.04 | 19.37 | - | 3858 | 3317 | - | - | - | - | 54.0 | 2.4 | 43.0 |
| MG_7.40-7.50 | Mistelgau | Core | 4461603 | 5529789 | 7.46 | 2.27 | - | 11.18 | 17.26 | - | 4025 | 3493 | - | - | - | 0.40 | 50.0 | 7.4 | 41.9 |





| Sample ID | Location | Sample type | Coordinates X | Coordinates Y | TVD m | $\rho_{b,dry}$ g/cm³ | $\rho_t$ g/cm³ | $\Phi_{Hg}$ % | $\Phi_{calc}$ % | $V_p$ m/s | $V_{p\,calc-Hg}$ m/s | $V_{p\,calc}$ m/s | GSF <2 μm % | GSF 2-63 μm % | GSF >63 μm % | VR %Ro | XRD Clay Minerals wt.-% | XRD $CaCO_3$ wt.-% | XRD Acc. min. wt.-% |
|---|---|---|---|---|---|---|---|---|---|---|---|---|---|---|---|---|---|---|---|
| MG_7.60-7.70 | Mistelgau | Core | 4461603 | 5529789 | 7.62 | 2.27 | 2.72 | 12.12 | 17.23 | - | 3940 | 3496 | - | - | - | - | - | - | - |
| MG_7.70-7.90 | Mistelgau | Core | 4461603 | 5529789 | 7.80 | 2.02 | 2.82 | 15.31 | 26.23 | - | 3659 | 2776 | - | - | - | - | - | - | - |
| DB_3.50 | Dörtbach | Core | 4453559 | 5468616 | 3.50 | 2.25 | - | 11.63 | 18.10 | - | 3984 | 3422 | - | - | - | 0.53 | 49.4 | 3.2 | 46.1 |
| DB_4.00 | Dörtbach | Core | 4453559 | 5468616 | 4.00 | - | - | - | - | - | - | - | - | - | - | - | 9.1 | 79.1 | 11.2 |
| DB_4.35 | Dörtbach | Core | 4453559 | 5468616 | 4.37 | 2.17 | 2.80 | 12.56 | 20.76 | - | 3901 | 3204 | - | - | - | - | 43.4 | 3.1 | 52.8 |
| DB_4.50 | Dörtbach | Core | 4453559 | 5468616 | 4.53 | 2.14 | 2.71 | 12.40 | 22.13 | - | 3915 | 3094 | - | - | - | - | 42.9 | 13.0 | 43.3 |
| DB_5.00 | Dörtbach | Core | 4453559 | 5468616 | 5.00 | 2.29 | - | 10.16 | 16.52 | - | 4118 | 3556 | - | - | - | - | 51.3 | 2.6 | 45.4 |
| DB_6.00-6.15 | Dörtbach | Core | 4453559 | 5468616 | 6.07 | 2.12 | 2.77 | 10.86 | 22.81 | - | 4054 | 3040 | - | - | - | - | 49.2 | 5.7 | 45.4 |
| DB_6.15-6.30 | Dörtbach | Core | 4453559 | 5468616 | 6.18 | - | - | - | - | - | - | - | - | - | - | - | 48.8 | 5.7 | 44.9 |
| DB_6.30-6.60 | Dörtbach | Core | 4453559 | 5468616 | 6.42 | - | - | - | - | - | - | - | - | - | - | - | 51.0 | 3.7 | 44.4 |
| DB_6.90-7.00 | Dörtbach | Core | 4453559 | 5468616 | 6.96 | - | - | - | - | - | - | - | - | - | - | - | 54.8 | 3.0 | 41.8 |
| DB_7.00-7.20 | Dörtbach | Core | 4453559 | 5468616 | 7.10 | - | - | - | - | - | - | - | - | - | - | - | 26.0 | 40.2 | 32.7 |
| DB_7.25-7.35 | Dörtbach | Core | 4453559 | 5468616 | 7.33 | - | - | - | - | - | - | - | - | - | - | 0.74 | 51.4 | 3.5 | 44.5 |
| DB_7.90-8.00 | Dörtbach | Core | 4453559 | 5468616 | 7.95 | - | - | - | - | - | - | - | - | - | - | - | 52.4 | 3.0 | 44.2 |
| DB_8.10-8.25 | Dörtbach | Core | 4453559 | 5468616 | 8.13 | - | - | - | - | - | - | - | - | - | - | - | 54.2 | 3.1 | 42.1 |
| DB_8.75-8.90 | Dörtbach | Core | 4453559 | 5468616 | 8.80 | - | - | - | - | - | - | - | - | - | - | - | 53.4 | 3.4 | 42.4 |
| DB_9.00-9.15 | Dörtbach | Core | 4453559 | 5468616 | 9.05 | - | - | - | - | - | - | - | - | - | - | - | 51.9 | 3.3 | 44.3 |
| DB_9.25-9.35 | Dörtbach | Core | 4453559 | 5468616 | 9.30 | - | - | - | - | - | - | - | - | - | - | - | 51.8 | 4.2 | 43.4 |
| DB_9.55-9.70 | Dörtbach | Core | 4453559 | 5468616 | 9.63 | - | - | - | - | - | - | - | - | - | - | - | 53.9 | 3.9 | 41.6 |
| DB_10.00 | Dörtbach | Core | 4453559 | 5468616 | 10.02 | - | - | - | - | - | - | - | - | - | - | - | 51.9 | 3.3 | 44.5 |
| DB_10.35 | Dörtbach | Core | 4453559 | 5468616 | 10.35 | - | - | - | - | - | - | - | - | - | - | - | 53.2 | 3.0 | 43.2 |
| DB_10.45 | Dörtbach | Core | 4453559 | 5468616 | 10.45 | - | - | - | - | - | - | - | - | - | - | - | 52.8 | 4.0 | 42.8 |
| Itt_V87_20.08 | Itting | Core | 4453389 | 5498496 | 20.08 | 2.25 | 2.69 | 11.25 | 17.96 | - | 4019 | 3492 | - | - | - | - | - | - | - |



| Sample ID | Location | Sample type | Coordinates X | Coordinates Y | TVD m | ρb,dry g/cm³ | ρt g/cm³ | ΦHg % | Φcalc % | Vp m/s | Vpcalc-Hg m/s | Vpcalc m/s | GSF <2 µm % | GSF 2-63 µm % | GSF >63 µm % | VR %Ro | XRD Clay Minerals wt.-% | XRD CaCO₃ wt.-% | XRD Acc. min. wt.-% |
|---|---|---|---|---|---|---|---|---|---|---|---|---|---|---|---|---|---|---|---|
| Itt_V87_20.45 | Itting | Core | 4455389 | 5498496 | 20.45 | - | - | - | - | - | - | - | - | - | - | - | 34.3 | 31.3 | 33.8 |
| Itt_V87_21.00 | Itting | Core | 4455389 | 5498496 | 21.00 | 2.27 | 2.69 | 10.18 | 17.27 | - | 4116 | - | - | - | - | - | 37.4 | 7.7 | 54.5 |
| Itt_V87_21.04 | Itting | Core | 4455389 | 5498496 | 21.04 | 2.28 | 2.65 | 9.84 | 16.95 | - | 4147 | 3519 | - | - | - | - | 33.4 | 7.8 | 58.0 |
| Itt_V87_21.38 | Itting | Core | 4455389 | 5498496 | 21.38 | - | - | - | - | - | - | - | - | - | - | - | 42.4 | 10.7 | 46.0 |
| Itt_V87_21.51 | Itting | Core | 4455389 | 5498496 | 21.51 | 2.33 | 2.71 | 10.47 | 15.15 | - | 4090 | - | - | - | - | - | - | - | - |
| Itt_V87_21.80 | Itting | Core | 4455389 | 5498496 | 21.80 | - | - | - | - | - | - | - | - | - | - | - | 41.6 | 9.5 | 48.6 |
| B05 | Eichstätt | B. g. | 4442241 | 5415335 | 327.00 | - | - | - | - | 2890 | - | - | - | - | - | - | - | - | - |
| B10 | Daiting | B. g. | 4418359 | 5406274 | 455.00 | - | - | - | - | 2650 | - | - | - | - | - | - | - | - | - |
| 4/85 | Zapfendorf | Sonic Log | 4429110 | 5542650 | 15.80 | - | - | - | - | 2890 | - | - | - | - | - | - | - | - | - |
| 4/85 | Zapfendorf | Sonic Log | 4429110 | 5542650 | 16.50 | - | - | - | - | 2725 | - | - | - | - | - | - | - | - | - |
| 4/85 | Zapfendorf | Sonic Log | 4429110 | 5542650 | 17.00 | - | - | - | - | 2681 | - | - | - | - | - | - | - | - | - |
| 4/85 | Zapfendorf | Sonic Log | 4429110 | 5542650 | 18.80 | - | - | - | - | 2770 | - | - | - | - | - | - | - | - | - |
| 4/85 | Zapfendorf | Sonic Log | 4429110 | 5542650 | 19.80 | - | - | - | - | 2841 | - | - | - | - | - | - | - | - | - |
| 4/85 | Zapfendorf | Sonic Log | 4429110 | 5542650 | 20.90 | - | - | - | - | 2604 | - | - | - | - | - | - | - | - | - |
| 4/85 | Zapfendorf | Sonic Log | 4429110 | 5542650 | 21.60 | - | - | - | - | 3125 | - | - | - | - | - | - | - | - | - |
| 4/85 | Zapfendorf | Sonic Log | 4429110 | 5542650 | 22.10 | - | - | - | - | 3115 | - | - | - | - | - | - | - | - | - |
| 4/85 | Zapfendorf | Sonic Log | 4429110 | 5542650 | 22.50 | - | - | - | - | 3205 | - | - | - | - | - | - | - | - | - |
| 4/85 | Zapfendorf | Sonic Log | 4429110 | 5542650 | 23.10 | - | - | - | - | 3356 | - | - | - | - | - | - | - | - | - |
| 4/85 | Zapfendorf | Sonic Log | 4429110 | 5542650 | 24.10 | - | - | - | - | 3300 | - | - | - | - | - | - | - | - | - |
| 4/85 | Zapfendorf | Sonic Log | 4429110 | 5542650 | 25.80 | - | - | - | - | 2584 | - | - | - | - | - | - | - | - | - |
| 4/85 | Zapfendorf | Sonic Log | 4429110 | 5542650 | 26.40 | - | - | - | - | 2525 | - | - | - | - | - | - | - | - | - |
| 4/85 | Zapfendorf | Sonic Log | 4429110 | 5542650 | 26.80 | - | - | - | - | 2500 | - | - | - | - | - | - | - | - | - |
| 4/85 | Zapfendorf | Sonic Log | 4429110 | 5542650 | 27.40 | - | - | - | - | 2433 | - | - | - | - | - | - | - | - | - |





| Sample ID | Location | Sample type | Coordinates | | TVD | $\rho_{b,dry}$ | $\rho_t$ | $\Phi_{Hg}$ | $\Phi_{calc}$ | Vp | $Vp_{calc-Hg}$ | $Vp_{calc}$ | GSF | | | VR | XRD | | |
|---|---|---|---|---|---|---|---|---|---|---|---|---|---|---|---|---|---|---|---|
| | | | X | Y | m | g/cm³ | g/cm³ | % | % | m/s | m/s | m/s | <2 μm % | 2-63 μm % | >63 μm % | %Ro | Clay Minerals wt.-% | CaCO₃ wt.-% | Acc. min. wt.-% |
| 4/85 | Zapfendorf | Sonic Log | 4429110 | 5542650 | 28.00 | - | - | - | - | 2392 | - | - | - | - | - | - | - | - | - |
| 4/85 | Zapfendorf | Sonic Log | 4429110 | 5542650 | 29.00 | - | - | - | - | 2392 | - | - | - | - | - | - | - | - | - |
| 162 | N. s. a. | S. s. | 4462717 | 5526786 | 17.88 | - | - | - | - | 2433 | - | - | - | - | - | - | - | - | - |
| 158 | N. s. a. | S. s. | 4459517 | 5529194 | 20.68 | - | - | - | - | 3354 | - | - | - | - | - | - | - | - | - |
| 157 | N. s. a. | S. s. | 4458397 | 5529759 | 21.98 | - | - | - | - | 2883 | - | - | - | - | - | - | - | - | - |
| 155 | N. s. a. | S. s. | 4456657 | 5530941 | 87.41 | - | - | - | - | 2203 | - | - | - | - | - | - | - | - | - |
| 168 | N. s. a. | S. s. | 4467573 | 5523752 | 38.59 | - | - | - | - | 2215 | - | - | - | - | - | - | - | - | - |
| 80 | N. s. a. | S. s. | 4425179 | 5552622 | 22.56 | - | - | - | - | 2935 | - | - | - | - | - | - | - | - | - |
| 83 | N. s. a. | S. s. | 4427645 | 5554515 | 20.86 | - | - | - | - | 2665 | - | - | - | - | - | - | - | - | - |
| 82 | N. s. a. | S. s. | 4426750 | 5553694 | 22.62 | - | - | - | - | 2637 | - | - | - | - | - | - | - | - | - |
| 116 | N. s. a. | S. s. | 4423532 | 5552959 | 40.74 | - | - | - | - | 2493 | - | - | - | - | - | - | - | - | - |
| 78 | N. s. a. | S. s. | 4423448 | 5551656 | 21.63 | - | - | - | - | 2387 | - | - | - | - | - | - | - | - | - |
| 76 | N. s. a. | S. s. | 4421799 | 5550442 | 15.37 | - | - | - | - | 2689 | - | - | - | - | - | - | - | - | - |
| 204 | N. s. a. | S. s. | 4427396 | 5541869 | 19.09 | - | - | - | - | 2614 | - | - | - | - | - | - | - | - | - |
| 203 | N. s. a. | S. s. | 4426895 | 5541393 | 19.86 | - | - | - | - | 2292 | - | - | - | - | - | - | - | - | - |
| 224 | N. s. a. | S. s. | 4444097 | 5554516 | 39.15 | - | - | - | - | 2185 | - | - | - | - | - | - | - | - | - |
| 232 | N. s. a. | S. s. | 4450911 | 5559442 | 21.79 | - | - | - | - | 2724 | - | - | - | - | - | - | - | - | - |