# Peer review of "Reconstructing post-Jurassic overburden in Central Europe: New insights from mudstone compaction and thermal history analyses of the Franconian Alb, SE Germany"

_Solid Earth, 2022_

## Referee Comment (RC1)

Review to

**Freitag et al. : Reconstructing post-Jurassic overburden in Central Europe: New insights from mudstone compaction and thermal history analyses of the Franconian Alb, SE Germany.**

In summary: Well written, well explained and reliable data-set!

Comments (please find the same and some additional in the review pdf):

1. Introduction: in general a good overview, only some remarks

Line 37: In contrast to Freudenberger (2013), a separated Upper Permian to Triassic Franconian basin never existed – I had a lot of discussions with him about this theme. There is no evidence for separation from the Central European (Germanic basin) in the sense that the Thuringian forest already existed and even not in the sense of a subbasin. Thickness and facies reflect a subsidence axis extending from Franconia to central Thuringia and further to Saxony-Anhalt to Brandenburg (Stratigraphy von Deutschland XI: Röhling & Lepper 2013)

Line 40: The flooding occurred both from the Tethys and from the north; I suggest skipping of Tethys Ocean.

Line 56: citation - more important than Voigt et al. 2008 and 2021 is Kley and Voigt 2008, content: The main effect of the compression was not the removal of Cretaceous sediments (in maximum some hundred metres of  Lower Cretaceous and Cenomanian) but the formation basement uplifts with amounts of several thousand metres at localized faults (Franconian line, Pfahl fault) – please look to Ziegler, or best to the book Littke et al. 2008: Dynamics of complex intracontinental basins.

Line 59: The later domal (?) uplift affects the Central European crust in total, and has probably nothing to do with the formation of the alps. New results and a good overview was published:

von Eynatten, H., Kley, J., Dunkl, I., Hoffmann, V.-E., and Simon, A.: Late Cretaceous to Paleogene exhumation in central Europe – localized inversion vs. large-scale domal uplift, Solid Earth, 12, 935–958, https://doi.org/10.5194/se-12-935-2021, 2021.

Line 62: The textbooks of Meschede and Walter are not primary sources, they refer to published articles. Please remove.

Line 93: Incomplete, please consider the Wasserburg and Regensburg basin (Cenomanian to Campanian).

Line 95: Even the Regensburg Basin and the northern parts of the Wasserburg basin suffered uplift – Please compare the age of preserved Cretaceous sediments and the AFT-ages – uplift of the basement east of the Franconian line continued, resulting in the deepening of the marginal trough. Campanian to Maastrichtian was removed later.

Line 113: I noted that you know the study of von Eynatten et al. 2021. It should be mentioned a little bit earlier to avoid misunderstandings.

Line 123: a break in organisation of the text: I suggest, to include the subchapter in the chapter "data and methods"

Complete Chapter 2.3

2.3.1.-2.3.3 I am not very familiar with the geophysical approach. My question is how carbonate and quartz content influence the equations. Are the samples represented completely pure mudstones?

Concerning organisation: could you shift the mineralogical part to a position before you consider the density and velocity chapter? This would answer my question before.

Chapter 2.4.

Line 299-302: please explain, why you applied the mineralogy. The sentence should be moved to methods (compaction, density, velocity)

Chapter 35.

Very good discussion! Only some remarks.

Line 461: On which base Wall et al estimated such a high thermal gradient? In which time? How rapid decreased this to the recent values? If the high rates of 40°/km are related to the Eger Rift (what I assume), they can probably not explain the high maturity of Jurassic.

What is the recent regional heat flow aside the anomalies?

Fig. 10: please add localities Mistelgau and Mürsbach on the map

Line 490: Markus Wilmsen and Birgit Niebuhr made many detailed investigations in the Cretaceous of the "Danubian Cretaceous Basin" and in the Bodenwöhr Basin. It would be better to cite them instead of me. You can easily find at least 5 relevant publications.

Line 504: Franconian Alb area is a bit misleading, because no relevant tectonics occurred there. Here it is better to write "close to the Franconian line", because the controlling structure is the Thuringian forest (probably as graben in Late Jurassic to Lower Cretaceous times and/or the Late Cretaceous marginal trough in front of the uprising inversion structure that formed later.

Could you please add a general time-frame in which the sediments were removed? Peterek and Schröder give some interesting data (timing of volcanism, sediment remnants of Cretaceous and Neogene age on the Franconian Alb)

Chapter 4 (conclusions)

Is the reason for the discrepancy of velocity and density data only the distance to the surface (weathering, decompaction and water saturation)? The fact should be already considered in the discussion and a clear statement should be given in the conclusions.

Question: As the velocity and density data only depend on the thickness of overburden and the VR-data solely on temperature and time, could an estimation of the heatflow during burial be possible? Alternatively, a Petromod model would be helpful if you have a vertical section with VR-data.

That is all from my side. Congratulations to a comprehensive, well-written paper. I have to think about whether Late Cretaceous or Paleogene erosion caused the recent surface geology. However, possibly you can put things together, if you look closer to the AFT-data in Franconia (Buntsandstein, eroded pebbles within the Upper Cretaceous).

Thomas Voigt, Jena

---

## Author Comment (AC1)

**Reconstructing post-Jurassic overburden in Central Europe: New insights from mudstone compaction and thermal history analyses of the Franconian Alb, SE Germany**

Response to reviewer's comments

Reviewer Dr. Thomas Voigt:

*Interesting paper with a consistent, well discussed data-set. The conclusions could be extended to some more results concerning paleo-heatflow and timing and rates of exhumation (pin-pointing time-span by surface geology).*

*More questions (for my own understanding) and some suggestions for better organisation of the paper are in the text. It is not necessary to send the revised version again to me.*

**Authors response**

The authors thank the reviewer Dr. Thomas Voigt for the positive feedback and comments provided for our manuscript. Your suggestions helped in improving the content, readability and strengthen the interpretation. On your recommondation, we extended the discussion and conclusion with respect to the timing and rates of exhumation, where we compare and discuss published results to ours. However, our data did not allow for the estimation of paleo-heatflow, which needs to be investigated in upcoming studies.

**Responses to comments on the text**

| Reviewer #1 comments | Authors answers |
|---|---|
| Line 37-38: In contrast to Freudenberger (2013), a separated Upper Permian to Triassic Franconian basin never existed – I had a lot of discussions with him about this theme. There is no evidence for separation from the Central European (Germanic basin) in the sense that the Thuringian forest already existed and even not in the sense of a subbasin. Thickness and facies reflect a subsidence axis extending from Franconia to central Thuringia and further to Saxony-Anhalt to Brandenburg (Stratigraphy von Deutschland XI: Röhling & Lepper 2013) | Lines 37-38: Our sentence must have been misleading and was changed accordingly. |
| Line 40: The flooding occurred both from the Tethys and from the north; I suggest skipping of Tethys Ocean. | Line 40: Text modified accordingly |
| Line 56: citation - more important than Voigt et al. 2008 and 2021 is Kley and Voigt 2008, content: The main effect of the compression was not the removal of Cretaceous sediments (in maximum some hundred metres of Lower Cretaceous and Cenomanian) but the formation basement | Lines 56-58: Citation was added and text changed in order to emphasize this causal relationship. |

| | |
|---|---|
| uplifts with amounts of several thousand metres at localized faults (Franconian line, Pfahl fault) – please look to Ziegler, or best to the book Littke et al. 2008: Dynamics of complex intracontinental basins. | |
| Line 60: The later domal (?) uplift affects the Central European crust in total, and has probably nothing to do with the formation of the alps. New results and a good overview was published:

von Eynatten, H., Kley, J., Dunkl, I., Hoffmann, V.-E., and Simon, A.: Late Cretaceous to Paleogene exhumation in central Europe – localized inversion vs. large-scale domal uplift, Solid Earth, 12, 935–958, https://doi.org/10.5194/se-12-935-2021, 2021 | Lines 60-62: This sentence must have been misunderstood, as we did not link these two processes to each other but suggest that both together are responsible for the southward tilting of the South German Mesozoic strata.
Also, we now mention this reference already here. |
| Lines 63: The textbooks of Meschede and Walter are not primary sources, they refer to published articles. Please remove. | Lines 56, 63-64, 93, 111, 467: References are removed. |
| Line 94: Incomplete, please consider the Wasserburg and Regensburg basin (Cenomanian to Campanian) | Lines 94-97: Text modified and Fig. 3 adapted accordingly. |
| Line 94: Even the Regensburg Basin and the northern parts of the Wasserburg basin suffered uplift – Please compare the age of preserved Cretaceous sediments and the AFT-ages – uplift of the basement east of the Franconian line continued, resulting in the deepening of the marginal trough. Campanian to Maastrichtian was removed later. | Lines 94-100: Text modified |
| Line 118: I noted that you know the study of von Eynatten et al. 2021. It should be mentioned a little bit earlier to avoid misunderstandings. | Line 118: Study was added earlier in line 62. |
| Line 127: a break in organisation of the text: I suggest, to include the subchapter in the chapter "data and methods" | Line 127: We deleted this chapter here and moved the modified text to chapter "3 Methods" in line 179. |
| Complete Chapter 2.3: 2.3.1.-2.3.3 I am not very familiar with the geophysical approach. My question is how carbonate and quartz content influence the equations. Are the samples represented completely pure mudstones? Concerning organisation: could you shift the mineralogical part to a position before you consider the density and velocity chapter? This would answer my question before. | Complete Chapter 2.3: With the added modified text at the beginning of this chapter, we now hope to clarify these questions.
The mineralogical part was moved before the compaction, density, and velocity subchapters. |

| | |
|---|---|
| Line 305-308: please explain, why you applied the mineralogy. The sentence should be moved to methods (compaction, density, velocity) | Lines 305-308: An explanation is now given in the shifted and modified text in chapter "3 Methods". |
| Line 487: On which base Wall et al estimated such a high thermal gradient? In which time? How rapid decreased this to the recent values? If the high rates of 40°/km are related to the Eger Rift (what I assume), they can probably not explain the high maturity of Jurassic.

What is the recent regional heat flow aside the anomalies? | Line 487: We try to avoid explaining all these details as this is beyond the scope of our manuscript. Instead the reader is referred to de Wall et al.'s published paper and the methods & data section therein.
We also refer to a more recent study by Kämmlein et al. (2020) which is based on corrected borehole data and provides a good overview of thermal gradients and calculated heat flow values.
The recent regional heat flow varies between 65-85 mW/m² according to Čermák and Bodri (1991). This information was added to the text. (line 493) |
| Fig. 10: please add localities Mistelgau and Mürsbach on the map | Fig. 10: Locations were added to figure 10. |
| Lines 500-501: Markus Wilmsen and Birgit Niebuhr made many detailed investigations in the Cretaceous of the "Danubian Cretaceous Basin" and in the Bodenwöhr Basin. It would be better to cite them instead of me. You can easily find at least 5 relevant publications | Lines 500-501: Thank you for the advice, we added some interesting and suitable publications. |
| Line 514: Franconian Alb area is a bit misleading, because no relevant tectonics occurred there. Here it is better to write "close to the Franconian line", because the controlling structure is the Thuringian forest (probably as graben in Late Jurassic to Lower Cretaceous times and/or the Late Cretaceous marginal trough in front of the uprising inversion structure that formed later.

Could you please add a general time-frame in which the sediments were removed? Peterek and Schröder give some interesting data (timing of volcanism, sediment remnants of Cretaceous and Neogene age on the Franconian Alb) | Line 514: Text changed accordingly.

We added a section which treats and discusses a general time-frame of the sedimentation and erosion history in this area since the Jurassic. (lines 524-535). |
| Chapter 4 (conclusion): Is the reason for the discrepancy of velocity and density data only the distance to the surface (weathering, decompaction and water saturation)? | Chapter 4 (conclusion): For the most part, yes. Of course also variation in mineralogy and texture influence the compaction state |

| | |
|---|---|
| The fact should be already considered in the discussion and a clear statement should be given in the conclusions.

Question: As the velocity and density data only depend on the thickness of overburden and the VRdata solely on temperature and time, could an estimation of the heatflow during burial be possible? Alternatively, a Petromod model would be helpful if you have a vertical section with VR-data | of mudstones, however, the main reason for the discrepancy results from different sensitivities of the applied methods to the stated factors. We added a clarifying sentence in lines 369-371 and give a clear statement in the conclusion on what methods are best suitable (lines 557-559).

To your question: Yes, that would be possible. However, the scarcity in VR-data and the fact that also the thermal conductivity of the nowadays removed Cretaceous sediments (which are thought to consitute the large majority of the overlying sediments according to our findings) would have to be estimated, we thinkt that a heat flow estimation during burial would be highly uncertain and would also be beyond the scope of this study. Furthermore, we did not have a vertical VR-data section, hence a Petromod model could not be calculated. Nevertheless, we will consider your question in future studies in this area and hope to give you a profound and more satisfying answer then. |

---

## Author Comment (AC2)

**Reconstructing post-Jurassic overburden in Central Europe: New insights from mudstone compaction and thermal history analyses of the Franconian Alb, SE Germany**

Response to reviewer's comments

Reviewer Prof. Dr. Hilmar von Eynatten:

*Simon Freitag and co-authors use petrographic and petrophysical properties and organic maturation data of Lower and Middle Jurassic mudstones from outcrops and drillcores of the Franconian Alb to estimate thicknesses of the post-Jurassic regional overburden. The paper is overall well written, methods and calibrations appear sound to me (though I'm not an expert in petrophysical properties), and the results constitute a significant and highly relevant contribution for the understanding of the Mesozoic evolution of the area. I recommend minor revisions only. The authors may consider separating chapter 3 into 'Results' (largely sections 3.1 to 3.4) and 'Discussion' (largely 3.5 and 3.6, could then be a new chapter 4).*

*When comparing the results to those by von Eynatten et al. (2021) in section 3.5, please consider that their modeling leading to 3-4 km burial refers to Early Triassic (Bundsandstein) strata (their figure 10). Including about 600-800 m of Middle Triassic (Muschelkalk) and Late Triassic (Keuper) strata significantly reduces the contrast between the two studies. Moreover, the study area is located towards the eastern/southern margin of the domal uplift proposed by von Eynatten et al. (2021) with likely less uplift/exhumation, as already emphasized in section 3.6. Given that the thermal anomalies mentioned are mainly local (as already stated by Freitag et al.) and an elevated heat flow of 80-85 mWm-2 still requires removal of 2.5-3 km of post-Early Triassic overburden (von Eynatten et al. 2021), I guess the contrast between the two studies remains within the uncertainties of the individual methods, implying that there is no need to call for increased heat flows or geothermal gradients.*

*Some parts appear over-referenced (and in this respect redundant in the Introduction and Methods parts, e.g. lines 125-126, 127-129, 133-134, 188-190). Please consider reducing to two or three major references as examples (e.g., …) or being more specific regarding information and respective references.*

**Authors response**

The authors thank the reviewer Prof. Dr. Hilmar von Eynatten for the constructive comments that helped in improving the content and quality of our manuscript. As recommended, we separated chapter 3 into 'Results' and 'Discussion', which contributed to a better structured and therefore more comprehensible manuscript. Reducing the amount of references in the over-referenced sections additionally increased the clarity of this manuscript. All the comments on the text have been addressed and reported in the table below.

**Responses to comments on the text**

| Reviewer #2 comments | Authors answers |
|---|---|
| Lines 80-82: sentence should be reformulated. | Lines 81-83: Sentence was reformulated. |

| | |
|---|---|
| Line 101: the Cretaceous strata are even more related to the parallel structure further south, not labelled in figure 1 but abbreviated as 'DF' in the inset (Bayrischer Pfahl?, not explained in caption). This should be clarified for readers not familiar with the regional geology. | Lines 99-104: This information was added to the sentence and figure 1 including caption modified. |
| Line 281: it remains unclear whether these are 41 individual samples or 41 measurements on ca. 10 samples (please note that in the heading for table 1 and in the text (line 157) the numbers summing up to 41 (in case of GSC) are declared as measurements per sample). The same holds for line 222: 72 samples (or measurements per sample?) for bulk density and porosity. This should be consistent and clear for the readers without checking the Appendix. | Line 281: Yes, there must have happened a mistake. We changed the heading of table 1 and it should now be clear that the numbers are equal to the number of samples, which had been analysed (one measurement per sample). |
| Line 286: these terms should be used in figure 4a as well (i.e. avoid clayshale, mudshale, siltshale, they are rather unusual). | Line 286: Figure 4 changed accordingly. |
| Line 314: quartz, pyrite, … | Line 314: Text modified. |
| Line 331: … (2018) suggests vertical effective stresses … … and roughly equates to 700-2000 m true vertical depth. | Lines 331-32: Text changed accordingly. |
| Line 440: just for consistency, lower limit is 800 m in Fig.9, caption to Fig. 9 and in the text (line 450). | Line 440: Text changed accordingly. |
| Line 472: not fully clear how the 1.1 km are deduced. | Lines 471-472: Text modified so that the origin of the 1.1 km should be clear now. |
| Line 490: von Eynatten et al. … | Line 490: Text changed accordingly. |
| Line 645: though correct for German name rules, 'von …' is usually listed under 'v' in the reference lists of international journals. The same holds for 'Le Bayon et al.', etc. I guess. | Line 827: Order in references changed accordingly. |